# Enteric glia regulate Paneth cell secretion and intestinal microbial ecology

Aleksandra Prochera[1†], Anoohya N Muppirala[1†], Gavin A Kuziel[1,2,3‡], Salima Soualhi[1‡], Amy Shepherd[1‡], Liang Sun[4], Biju Issac[4], Harry J Rosenberg[1,5], Farah Karim[6], Kristina Perez[1], Kyle H Smith[7], Tonora H Archibald[4], Seth Rakoff-Nahoum[1,2,3], Susan J Hagen[7], Meenakshi Rao[1]*

[1]Division of Gastroenterology, Department of Pediatrics, Boston Children's Hospital and Harvard Medical School, Boston, United States; [2]Division of Infectious Diseases, Department of Pediatrics, Boston Children's Hospital and Harvard Medical School, Boston, United States; [3]Department of Microbiology, Harvard Medical School, Boston, United States; [4]Research Computing, Department of Information Technology, Boston Children's Hospital, Boston, United States; [5]Department of Pathology, Beth Israel Deaconess Medical Center, Boston, United States; [6]Institute of Human Nutrition, Columbia University Irving Medical Center, New York, United States; [7]Department of Surgery, Beth Israel Deaconess Medical Center, Boston, United States

*For correspondence:
meenakshi.rao@childrens.harvard.edu

[†]These authors contributed equally to this work
[‡]These authors also contributed equally to this work

Competing interest: The authors declare that no competing interests exist.

## eLife Assessment

This study presents **important** findings on the function of enteric glia expressing proteolipid protein 1 (PLP1+ glia). The evidence supporting the claims of the authors is **solid**, although the inclusion of additional data showing the mechanisms by which PLP1+ enteric glia acts on Paneth cells would have strengthened the study. The work will be of interest to colleagues studying intestinal biology.

**Abstract** Glial cells of the enteric nervous system (ENS) interact closely with the intestinal epithelium and secrete signals that influence epithelial cell proliferation and barrier formation in vitro. Whether these interactions are important in vivo, however, is unclear because previous studies reached conflicting conclusions (Prochera and Rao, 2023). To better define the roles of enteric glia in steady state regulation of the intestinal epithelium, we characterized the glia in closest proximity to epithelial cells and found that the majority express the gene Proteolipid protein 1 (*PLP1*) in both mice and humans. To test their functions using an unbiased approach, we genetically depleted PLP1+ cells in mice and transcriptionally profiled the small and large intestines. Surprisingly, glial loss had minimal effects on transcriptional programs and the few identified changes varied along the gastrointestinal tract. In the ileum, where enteric glia had been considered most essential for epithelial integrity, glial depletion did not drastically alter epithelial gene expression but caused a modest enrichment in signatures of Paneth cells, a secretory cell type important for innate immunity. In the absence of PLP1+ glia, Paneth cell number was intact, but a subset appeared abnormal with irregular and heterogenous cytoplasmic granules, suggesting a secretory deficit. Consistent with this possibility, ileal explants from glial-depleted mice secreted less functional lysozyme than controls with corresponding effects on fecal microbial composition. Collectively, these data suggest that enteric glia do not exert broad effects on the intestinal epithelium but have an essential role in regulating Paneth cell function and gut microbial ecology.

## Introduction

The intestinal epithelium is an important interface between an animal and its external environment, not just as a physical barrier but also as a dynamic regulator of digestion, energy balance, and mucosal immunity. The ENS, the intrinsic nervous system of the digestive tract, directs many intestinal epithelial functions. Glia are major cellular components of the ENS distributed throughout the radial axis of the intestine, from the muscular outer walls to the inner mucosal layer containing the epithelium. Most studies of enteric glia have focused on the cells which closely associate with neuronal soma in the myenteric plexus and have uncovered numerous roles for these glia in the regulation of neuronal functions in both health and disease (*Seguella and Gulbransen, 2021*; *Rosenberg and Rao, 2021*). Many enteric glia, however, are located outside the myenteric plexus in the mucosa where they closely appose intestinal epithelial cells (*Ferri et al., 1982*; *Mestres et al., 1992*; *Krammer and Kühnel, 1993*; *Neunlist et al., 2007*; *Bohórquez et al., 2014*), raising the possibility that mucosal glia directly regulate epithelial cell functions.

Several enteric glia-derived cues, ranging from small molecules to growth factors, can alter epithelial cell proliferation and cell-cell adhesion in vitro (reviewed in *Prochera and Rao, 2023*), supporting the possibility of glial-epithelial interactions. In the intestine, however, none of these factors are made exclusively by glia (*Prochera and Rao, 2023*). Moreover, studies in which enteric glia were depleted or disrupted in vivo have reported conflicting findings in terms of epithelial effects. For example, chemical gliotoxins do not cause major epithelial deficits (*Aikawa and Suzuki, 1985*; *Nasser et al., 2006*). In contrast, a chemical-genetic model using a human *GFAP* promoter fragment to target glia in mice showed profound epithelial barrier defects and fulminant inflammation specifically in the distal small intestine (*Bush et al., 1998*). Subsequent studies by multiple groups using more targeted systems to deplete or functionally disrupt glia defined by *Plp1*, *Sox10*, or *Gfap* expression, however, have not supported this finding. They found no major defects in epithelial properties at steady state or increased vulnerability to inflammation upon glial disruption even though all these models exhibited deficits in other ENS-regulated functions, such as motility (*Rao et al., 2017*; *Yuan et al., 2020*; *Kovler et al., 2021*; *Baghdadi et al., 2022*). One study reported that simultaneous depletion of *Plp1*- and *Gfap*-expressing populations could provoke intestinal inflammation, implicating a subset of glia with particularly high *Gfap* transcript expression in the regulation of epithelial turnover through the secretion of Wnt proteins (*Baghdadi et al., 2022*). Genetic disruption of Wnt secretion in *Gfap*⁺ cells, however, affected epithelial turnover only upon radiation injury (*Baghdadi et al., 2022*). Thus, it largely remains unclear what, if any, epithelial functions enteric glia are necessary for in vivo at steady state.

To better delineate the functional significance of enteric glial-epithelial interactions in homeostasis, first we characterized the molecular phenotype of mucosal glia along the crypt-villus axis in both mice and humans. We found that *Plp1* was the most widely expressed marker of mucosal glia in both species, supporting the use of its promoter to probe glial functions in vivo. Then, to interrogate glial functions in an unbiased way, we depleted *Plp1*⁺ cells in mice and examined gene expression along the longitudinal axis of the intestine. Surprisingly, glial loss had minimal effects on the intestinal transcriptome or the cellular composition of the epithelium. Targeting *Plp1*⁺ cells, however, caused a specific defect in Paneth cells leading to enriched gene expression signatures, diminished antimicrobial peptide secretion, and altered gut microbial composition. These observations uncover a link between enteric glia and Paneth cells and establish a role for enteric glia in regulating epithelial function in the healthy intestine in vivo.

## Results

### Plp1 expression broadly marks glia in the gut mucosa

Enteric glia, like Schwann cells in the periphery, are neural crest-derived and have been identified within tissues by molecular markers including *Sox10*, *Gfap*, *Plp1*, and *S100b*. Although most glia within enteric ganglia are labeled by these markers, heterogeneity in their expression has been reported (*Jessen and Mirsky, 1983*; *Boesmans et al., 2015*), particularly in glia located outside the myenteric plexus (*Rao et al., 2015*). To determine which marker is most broadly expressed by mucosal glia and would thus be most useful for genetic interrogation of glial-epithelial interactions,

we analyzed publicly available single-cell RNA sequencing (scRNAseq) datasets of human and mouse intestinal mucosa.

In human small and large intestines, expression of *SOX10* and *PLP1* was restricted to cluster of cells with a glial signature (*Figure 1A*, *Figure 1—figure supplement 1A, B*). *S100B* was highly expressed by cells in this cluster but also detected in non-glial cells including macrophages and monocytes; *GFAP* was overall undetectable (*Figure 1A*, *Figure 1—figure supplement 1A, B*). To validate these findings, we examined expression of the corresponding proteins by immunohistochemistry (IHC) in the human small intestine and found SOX10-, PLP1-, and S100B-immunoreactive cells in three compartments along the radial axis: the mucosa, enteric ganglia in the submucosal and myenteric plexuses, as well as intramuscular glia in the muscularis externa (*Figure 1—figure supplement 2A, B, D*). GFAP-immunoreactive cells were readily found within enteric ganglia but were rare in the mucosa (*Figure 1—figure supplement 2C*). These observations are consistent with a recent study that examined SOX10, GFAP, and S100B by IHC in the human colon and found little to no GFAP, but robust SOX10 and S100B expression across all three compartments (*Baidoo et al., 2023*). In the small intestine of patients with Crohn's disease, a type of inflammatory bowel disease (IBD), *SOX10*, *PLP1*, and *S100B* transcripts were highly enriched in mucosal glia while *GFAP* was enriched in various non-glial cells including fibroblasts and immune cells (*Figure 1B*). In sum, *SOX10, PLP1, and S100B* are broadly expressed by human enteric glia across the radial axis of the gut, including mucosal glia, in both healthy and inflamed states, with *SOX10* and *PLP1* exhibiting the most cell type specificity.

In mice, we previously showed that PLP1 is widely expressed by enteric glia across the radial axis of the small and large intestines where its expression largely overlaps with S100B; a more limited subset expresses GFAP (*Rao et al., 2015*). Consistent with these observations, analysis of scRNAseq data from the mouse colonic mucosa showed that *Plp1* is broadly expressed across the single putative cluster of mucosal glia; *Gfap*, *S100b*, and *Sox10* are also detectable in this cluster to variable extents (*Figure 1—figure supplement 1C*). Given recent observations that a subset of Gfap$^{high}$/Plp1$^{low}$ mucosal glia might be particularly important for epithelial regulation in the mouse small intestine (*Baghdadi et al., 2022*), we closely compared *Plp1* and *Gfap* expression in the mouse ileum. We performed whole-mount IHC for GFAP in Plp1-eGFP reporter mice to ensure the detection of colocalization despite any potential differences in subcellular distribution. Previous work from our lab and others has validated that eGFP expression in this reporter strain faithfully mirrors endogenous PLP1 expression within the enteric and central nervous systems (*Rao et al., 2015*; *Mallon et al., 2002*). In adult Plp1-eGFP mice, we found that mucosal glia diverged sharply in terms of marker expression based on their location along the crypt-villus axis. While the majority of villus *Plp1*$^+$ cells co-expressed GFAP, virtually none of the crypt-associated *Plp1*$^+$ cells did so (*Figure 1C and D*). With rare exceptions in some villi, GFAP-immunoreactive cells that did not express *Plp1* were largely undetectable. Together, these observations indicate that *Plp1* expression broadly marks enteric glia in mouse and human tissues and is among the most sensitive and specific markers for glia in the gut mucosa.

## Genetic depletion of enteric glia causes muted and region-specific changes in the intestinal transcriptome

To determine what aspects of intestinal homeostasis enteric glia are essential for in vivo, we took an unbiased approach. We examined changes in gene expression that occur upon glial loss in three different intestinal regions: proximal small intestine (duodenum), distal small intestine (ileum), and the large intestine (colon). We depleted the cells by administration of tamoxifen to young adult Plp1$^{CreER}$ Rosa26$^{DTA/+}$ mice, which we previously showed exhibit loss of the majority of S100B- and SOX10-expressing enteric glia, including 90% of mucosal glia (*Rao et al., 2017*). In this model, glia are lost within 5 days of tamoxifen induction (5dpt) and remain stably depleted through 14dpt; notably female, but not male, mice have intestinal dysmotility (*Rao et al., 2017*). To facilitate detection of direct effects of glia rather than indirect effects related to dysmotility, we isolated intestinal segments from male Plp1$^{CreER}$ Rosa26$^{DTA/+}$ mice and Cre-negative Rosa26$^{DTA/+}$ littermate controls at 11dpt and performed bulk RNA-sequencing (*Figure 2A*). Surprisingly, there were minimal changes observed in the transcriptome in all three regions of the intestine though the mice exhibited robust depletion of S100B$^+$ glia by IHC. In the duodenum and ileum, differential gene expression analysis by DESeq2 revealed no genes that reached the standard threshold for statistical significance of padj <0.05 (*Figure 2B*, *Figure 2—figure supplement 1*). In the colon, only five genes were differentially expressed, most

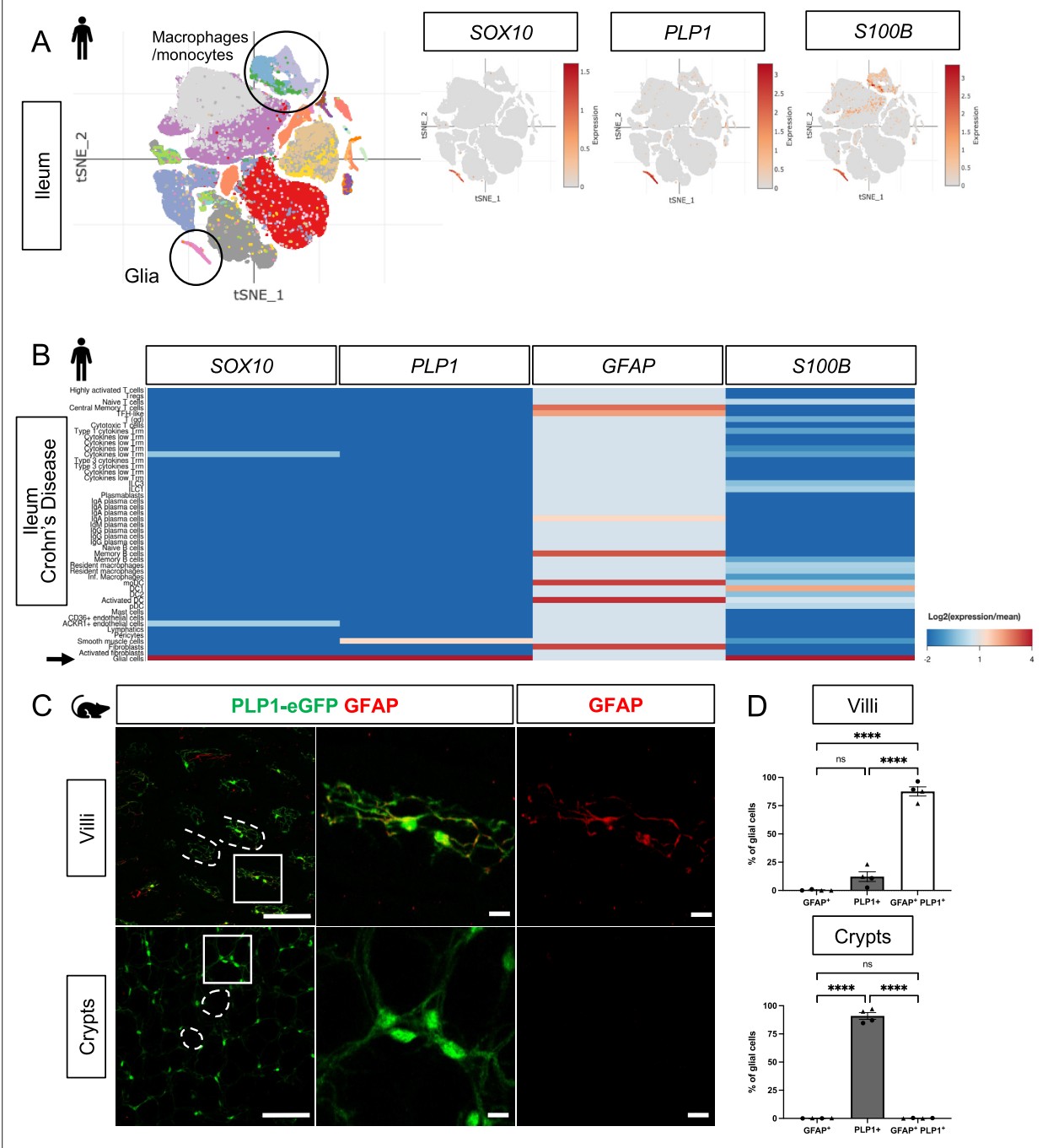

**Figure 1.** Mucosal glia in human and mouse small intestines widely express *Plp1*. (**A**) t-SNE plot of 94,451 cells isolated from terminal ileal mucosal biopsies from 13 children with non-inflammatory, functional gastrointestinal disorders (*Zheng et al., 2023*), colored by annotated cell identity. *PLP1* and *SOX10* expression exhibit relative specificity to glia; the cells express high levels of these transcripts. *S100B* is expressed by glia as well as non-glial cells such as macrophages and monocytes. *GFAP* is undetectable in this dataset. (**B**) Heatmap of gene expression from 82,417 cells obtained by scRNAseq of mucosal biopsies from inflamed and non-inflamed segments of terminal ileum obtained from 11 adults with Crohn's disease (*Martin et al., 2019*). In contrast to *PLP1*, *SOX10*, and *S100B*, which are most highly expressed in glia (arrow), *GFAP* expression is highest in non-glial cells. (**C**) Whole-mount immunostaining of ileum from an adult Plp1-eGFP mouse for GFAP imaged at the level of the villus- (top panels) and crypt-associated mucosa (bottom panels). Most glia in the villi express both *Plp1* and GFAP while virtually all glia in the mucosa surrounding epithelial crypts are *Plp1*+ and not immunoreactive for GFAP. (**D**) Quantification of the percentages of GFAP+, PLP1+, GFAP+ PLP1+ cells in the mucosa. Each data point represents an individual mouse, with triangles representing males and circles representing females (n=4). Scale bars = 100 μm (large panels) and 20 μm (magnified images). Error bars represent SEM. **** p<0.0001 by one-way ANOVA with Tukey multiple comparisons test.

The online version of this article includes the following figure supplement(s) for figure 1:

*Figure 1 continued on next page*

Figure 1 continued

**Figure supplement 1.** *Plp1* is the most specific and widely expressed marker of enteric glia in human and mouse colonic mucosa.

**Figure supplement 2.** S100B, SOX10, PLP1, and GFAP expression across the radial axis of the human small intestine.

of which were upregulated in glial-ablated mice (*Figure 2B*, *Figure 2—figure supplement 1*). These results suggest that acute depletion of enteric glia in male mice has limited effects on the intestinal transcriptome.

Enteric glia represent a relatively small proportion of cells in the intestine. Reasoning that transcriptional changes resulting from the biological effects of enteric glial loss might be muted in magnitude but consistent along the length of the intestine, we identified genes differentially expressed in Cre⁻ (controls) compared to Cre⁺ (glial-ablated) mice using a more lenient significance threshold of $p < 0.05$ and then performed DiVenn analysis (*Sun et al., 2019*) to identify changes that were shared across the duodenum, ileum, and colon. Most genes that were differentially expressed were highly specific to one region of the intestine (*Figure 2C*). While 331, 516, and 916 genes were changed uniquely in duodenum, ileum, and colon, respectively, only 16–29 genes were shared between pairs of tissues (*Figure 2C*). Remarkably, only two genes were differentially expressed between Cre⁻ and Cre⁺ mice in all three tissue regions (*Igkv4-91* and *Ighv1-58*), suggesting that enteric glia exert region-specific effects along the longitudinal axis of the intestine.

Focusing on the colon, which showed the most evidence of altered gene expression upon glial loss, we examined the expression of *Lyz1*, the top gene upregulated in Cre⁺ mice (*Figure 2B*, *Figure 2—figure supplement 1B*). LYZ1 is an antimicrobial peptide (AMP) that is highly and relatively specifically expressed by Paneth cells in the small intestine (*Figure 2—figure supplement 1A*; *Peeters and Vantrappen, 1975*). Quantitative RT-PCR (qPCR) of colonic tissue isolated from independent cohorts of Plp1^CreER Rosa26^DTA/+ mice confirmed upregulation of *Lyz1* in the colons of Cre⁺ mice (*Figure 2—figure supplement 2B*). Reactome pathway analysis of differentially enriched genes in the colon also highlighted pathways characteristic of Paneth cells including defensins and other AMPs (*Figure 2—figure supplement 2C*). Paneth cells are not typically present in the healthy mouse colon and their ectopic appearance is considered a marker of inflammation (*Hertzog, 1937*; *Paterson and Watson, 1961*; *Tanaka et al., 2001*; *Singh et al., 2020*). To determine if glial depletion provoked the formation of ectopic Paneth cells, we performed IHC for LYZ1 and DEFA5, a second and independent marker of Paneth cells (*Jones and Bevins, 1992*; *Salzman et al., 2003*). Although both markers robustly labeled Paneth cells in the small intestine, no LYZ1- or DEFA5-immunoreactive epithelial cells were detected in the colons of either Cre⁻ or Cre⁺ mice (*Figure 2—figure supplement 2D*). These data suggest that acute glial depletion causes transcriptional dysregulation in the colon linked to Paneth cell biology without evidence of ectopic Paneth cells or corresponding changes in proteins.

## Glial depletion selectively alters Paneth cells in the small intestine

Previous studies have indicated that glia might be most important for epithelial homeostasis in the ileum (*Bush et al., 1998*; *Cornet et al., 2001*). Although epithelial cells are well-represented in whole gut transcriptomes, there are many other abundant cell types such as immune cells that are also present. To investigate glial effects on epithelial cells more specifically, we mechanically isolated the ileal epithelium from glial-ablated and control mice at 11dpt and examined gene expression by RNA-Seq. DESeq2 and DiVenn analysis detected minimal overlap in the transcriptional changes observed in the whole ileum compared to the ileal epithelium, supporting the utility of focused epithelial analysis (*Figure 3A*).

Most intestinal epithelial cells turn over every 3–5 days (*Darwich et al., 2014*) and thus the majority of cells represented in the epithelial transcriptome of Cre⁺ mice would not have experienced glial interactions. Nevertheless, epithelial gene expression was similar in control and glial-ablated mice, mirroring the findings from whole tissue. No genes reached the $p_{adj} < 0.05$ threshold of significance for differential expression (*Figure 3—figure supplement 1A–C*).

The intestinal epithelium is composed of a diverse array of cells including absorptive enterocytes, Lgr5⁺ stem cells, and various secretory cell types (*Figure 3B*). To determine if glial depletion selectively affected any of these cell types, we performed gene set enrichment analysis (GSEA) using cell-type-specific signatures obtained from a published scRNAseq study (*Haber et al., 2017*; *Supplementary file 1*). Several of these signatures, most significantly that of Paneth cells ($p < 0.001$, FDR $< 0.001$), were

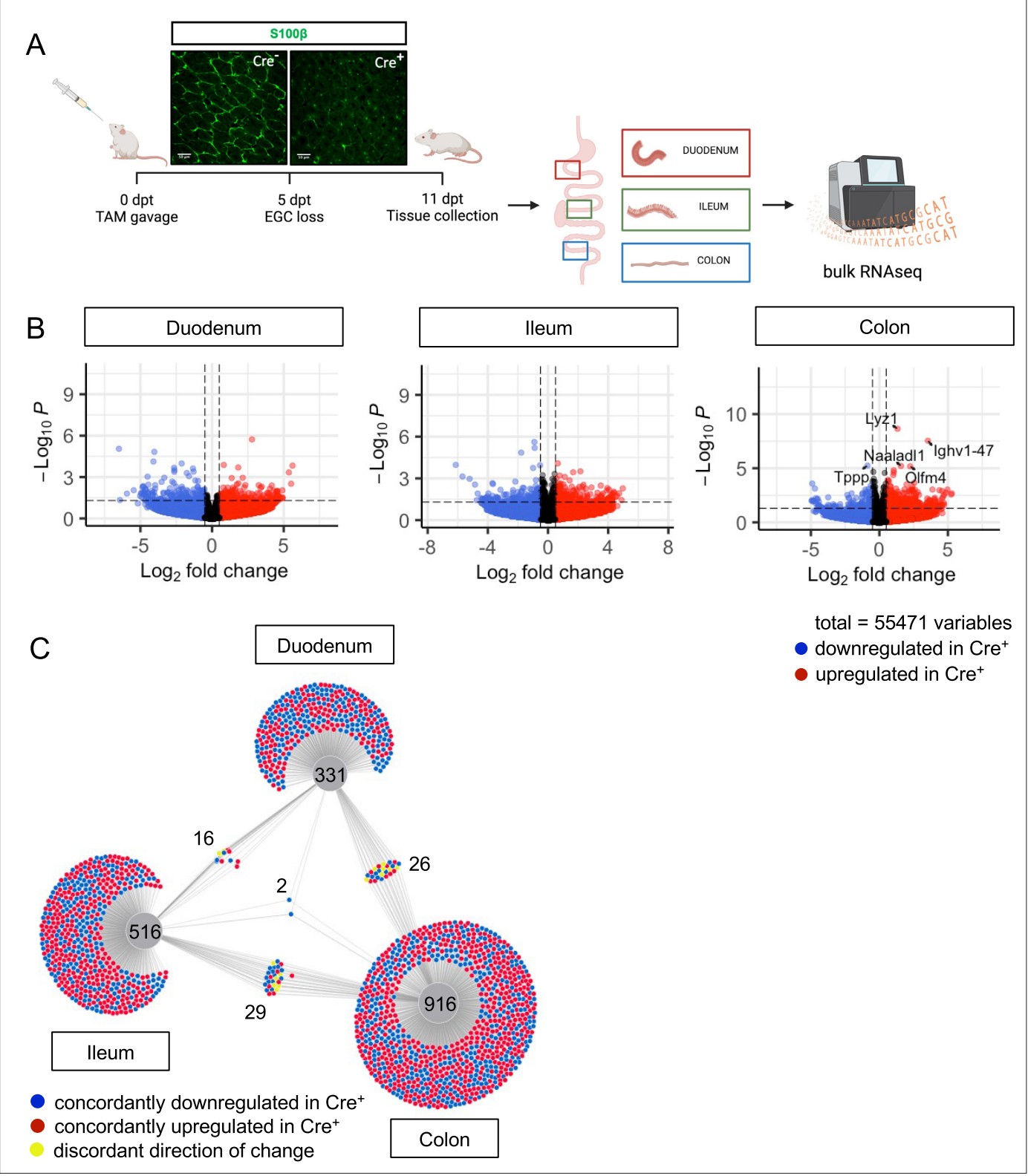

**Figure 2.** Glial ablation induces muted, region-specific transcriptional changes along the longitudinal axis of the intestine. (**A**) Schematic of the experimental timeline for bulk RNA-sequencing of intestinal tissue segments from male Plp1^CreER Rosa26^DTA/+ mice (annotated as Cre+) and Rosa26^DTA/+ littermate controls (annotated as Cre-). Tissues were collected 11 days after administration of tamoxifen (11dpt; n=4 per genotype). In this model, the majority of enteric glial cells (EGC) are eliminated by 5dpt, as seen on the representative IHC image of S100B staining in Cre+ and Cre- small intestinal

*Figure 2 continued on next page*

*Figure 2 continued*

mucosa. Glial depletion is stable through at least 14dpt (*Rao et al., 2017*). Panel A created with BioRender.com. (**B**) Volcano plots showing differentially expressed genes in duodenum, ileum, and colon of Cre- and Cre+ mice. Genes that reached statistical significance cutoff of *padj* <0.05 are labeled. Red and blue colors denote up- and down-regulated genes in Cre+ mice compared to Cre- mice with p-value <0.05, respectively. Differential analysis was conducted using DESeq2. (**C**) DiVenn analysis illustrates genes that were up- (red) or down-regulated (blue) in the duodenum, ileum, and colon of Cre+ mice compared to Cre- controls at 11dpt with p<0.05 threshold for significance. Nodes linking tissues constitute genes that were differentially expressed in Cre+ mice compared to Cre- controls in both of those tissues. Yellow color marks genes with discordant direction of change between the different tissue regions. Numbers indicate the number of genes at each node or tissue segment that were identified as differentially expressed. Overall, this analysis illustrates that most differentially expressed genes in Cre+ mice were region-specific with little overlap between duodenum, ileum, and colon.

The online version of this article includes the following figure supplement(s) for figure 2:

**Figure supplement 1.** Changes to whole tissue transcriptomes resulting from glial ablation.

**Figure supplement 2.** Glial ablation induces colonic expression of *Lyz1* at the transcript, but not protein, level.

enriched in the transcriptional profile of Cre[+] mice (*Figure 3B, Figure 3—figure supplement 1D*). An independent GSEA using curated cell signatures derived from bulk RNASeq studies also showed an enrichment of the Paneth cell program (p<0.001, FDR <0.001, *Figure 3—figure supplement 1E, F*).

The observed enrichment of Paneth or other secretory cell signatures could be a result of altered differentiation and/or survival. Immunostaining for molecular markers of Paneth, Lgr5[+], goblet, enteroendocrine, and microfold (M) cells, however, revealed no difference in their densities in Cre[+] mice compared to Cre[-] littermates (*Figure 3C–D*). In glial-ablated mice, all these cell types also appeared grossly normal, except for Paneth cells (*Figure 4A*). Paneth cells are highly secretory cells located at the crypt base that are responsible for production and release of the bulk of small intestinal AMPs, such as LYZ1 and α-defensins, which are crucial for homeostatic regulation of the microbiome and innate immunity (*Peeters and Vantrappen, 1975*; *Jones and Bevins, 1992*; *Salzman et al., 2003*; *Wilson et al., 1999*; *Salzman et al., 2010*; *Clevers and Bevins, 2013*). Labeling Paneth cell granules with the fucose-specific lectin UEA-1, revealed that many Paneth cells in Cre[+] mice had heterogenous secretory granules, some of which appeared giant, fused, or dysmorphic (*Figure 4A*). On ultrastructural analysis by transmission electron microscopy, Paneth cells in Cre[-] mice had typical morphology with a pyramidal shape, extensive rough endoplasmic reticulum, and relatively homogenous, electron-dense granules with haloes, which were oriented toward the apical surface of the cell (*Figure 4B*). In Cre[+] mice, Paneth cells had normal rough endoplasmic reticulum, but many exhibited a globular morphology and contained more heterogeneous granules (*Figure 4B*). In contrast, neighboring intestinal stem cells in the crypt base, as well as other secretory cell types such as enteroendocrine cells and goblet cells appeared no different in Cre[-] and Cre[+] mice (*Figure 4—figure supplement 1*). In sum, glial depletion did not provoke major shifts in small intestinal epithelial gene expression or cell composition but caused upregulation of Paneth cell genes associated with specific morphological changes that were highly specific to this cell type.

## Glial depletion impairs Paneth cell secretory activity

Paneth cells secrete their granules both constitutively and in response to various stimuli, such as pathogen-associated molecular patterns (*Ayabe et al., 2000*) and cholinergic agonists (*Satoh et al., 1989*). At the level of individual cells, disruption of this secretory activity can manifest as accumulation and/or fusion of their secretory granules (*Satoh, 1988*; *Ahonen, 1973*). The abnormal granule appearance in Cre[+] mice suggested that glial depletion might compromise Paneth cell secretion. Consistent with this possibility, the 'extracellular' and 'secretory' cellular compartments were most enriched in pathway analysis of Paneth cell genes that were changed in glial-depleted mice (*Figure 5A*). Paneth cell secretion has often been measured in preparations of mechanically isolated epithelial crypts or enteroids (*Ayabe et al., 2000*; *Yokoi et al., 2019*). These preparations, however, are denervated and lack key neighboring cells including glia. To enable measurement of Paneth cell secretion in a more native environment, we developed an explant-based activity assay to measure luminal lysozyme secretion (*Figure 5B*). Supporting this assay's specificity for Paneth cell-derived lysozyme, pre-treatment of mice with dithizone, a zinc chelator known to selectively deplete Paneth cell granules (*Sawada et al., 1991*; *Lueschow et al., 2018*), reduced detectable lysozyme activity (*Figure 5C*). Utilizing this assay, we found that small intestinal explants from Cre[+] mice secreted less active lysozyme than those from Cre[-] controls (*Figure 5C*), indicating that glial loss disrupts Paneth cell secretion.

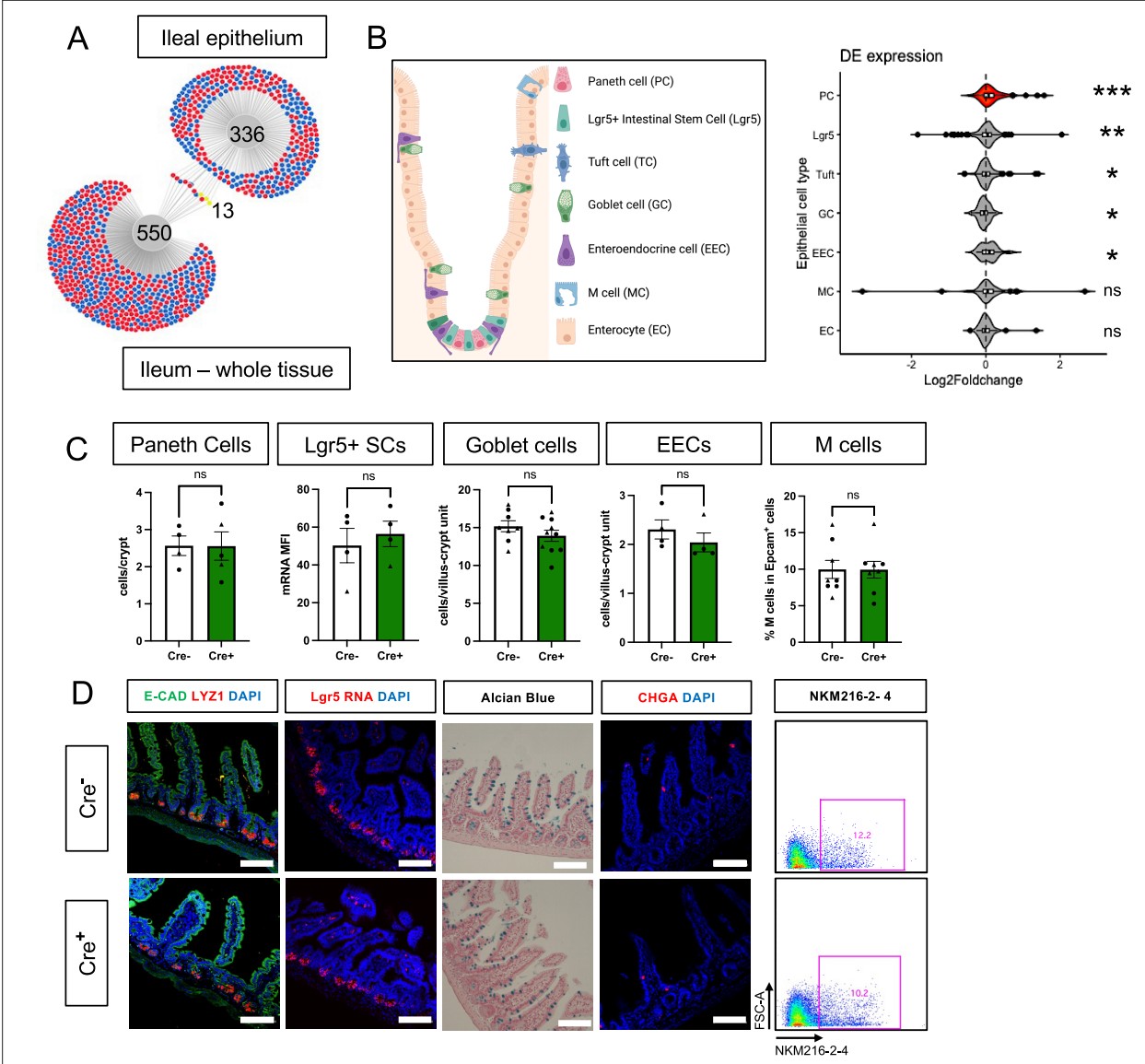

**Figure 3.** Glial ablation causes enrichment of specific epithelial cell type signatures without altering epithelial composition. (**A**) DiVenn analysis illustrates genes that were consistently up- (red) or down-regulated (blue) in the ileal epithelia and full-thickness ileal segments of Cre+ mice compared to Cre- controls at 11dpt with p<0.05 threshold for significance. Yellow color marks genes with discordant direction of change between the ileal epithelia and full-thickness ileal segments. (**B**) Gene set enrichment analysis (GSEA) of gene expression data from Cre+ vs. Cre- ileal epithelium using single-cell gene signatures for epithelial cell types (schematic representation of component cell types on the left) derived from *Haber et al., 2017*, *Supplementary file 1*. The Paneth cell signature was most significantly enriched in the ileal epithelium of glia-depleted mice. Red color denotes the significant enrichment consistent across two independent GSEA. Thresholds for DE analysis: p-value <0.05. *** p<0.001, FDR <0.001, ** p<0.001, FDR <0.01, * p<0.05, FDR <0.05, ns – non-significant. Panel B created with BioRender.com. (**C, D**) Quantification of epithelial subtypes in the small intestines of Cre- and Cre+ mice with representative IHC images and flow cytometry plots below each graph showing the marker and approach used for cell identification. Each data point represents an individual mouse, with triangles representing males and circles representing females (Paneth cells: n=4 for Cre-, n=5 for Cre+; Lgr5+ SCs: n=4 for Cre- and Cre+; Goblet Cells: n=8 for Cre-, n=10 for Cre+; EECs: n=4 for Cre- and Cre+, M cells: n=8 for Cre- and Cre+). Error bars represent SEM. ns - not significant by unpaired parametric *t*-test. Scale bar = 100 μm. E-Cadherin (E-CAD) labels cell borders, LYZ1 marks Paneth cells, *Lgr5* transcript expression marks intestinal stem cells (SCs), Alcian blue marks goblet cells, Chromogranin A (CHGA) marks enteroendocrine cells, and NKM216-2-4 identifies microfold (M) cells by flow cytometry. Cell nuclei are labeled with DAPI (blue) in the IHC panels.

The online version of this article includes the following figure supplement(s) for figure 3:

**Figure supplement 1.** Transcriptional profiling of the ileal epithelium in glial-ablated mice reveals enrichment of Paneth cell signatures.

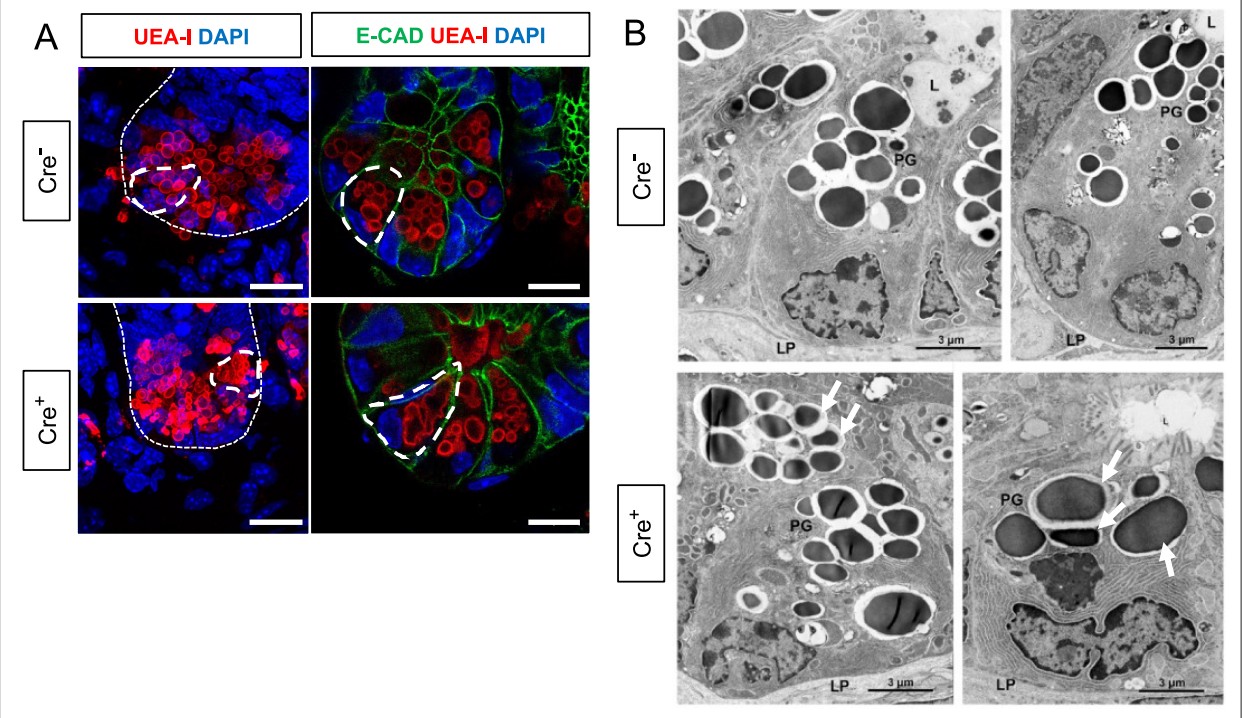

**Figure 4.** Glial depletion triggers morphological changes in Paneth cells. (**A**) Representative images of UEA-I staining of Paneth cell granules in the small intestine of Cre- and Cre+ mice (observed in at least three mice per genotype). Scale bar = 10 μm. (**B**) Representative transmission electron microscopy images of Paneth cells (n=2 mice per genotype from independent cohorts). Paneth cells in Cre+ mice are globular, exhibit loss of polarity, and have heterogeneous granules (arrows indicate errant granules). L, Lumen of the intestinal crypts; LP, lamina propria; PG, Paneth granule. Scale bar = 3 μm.

The online version of this article includes the following figure supplement(s) for figure 4:

**Figure supplement 1.** Glial depletion does not affect the ultrastructure of enteroendocrine cells, crypt base stem cells, or goblet cells.

Cholinergic signaling regulates Paneth cell function (*Satoh et al., 1989*; *Satoh, 1988*; *Satoh et al., 1992*; *Satoh et al., 1995*; *Dolan et al., 2022*) and genetic depletion of G proteins that act downstream of muscarinic acetylcholine receptors (AChR) alters granule morphology (*Watanabe et al., 2016*). Muscarinic acetylcholine receptor 3 (mAChR3) is the major neurotransmitter receptor expressed by Paneth cells (*Figure 5—figure supplement 1A, B*; *Dolan et al., 2022*). Its expression level in the epithelium was unchanged by glial loss (log2FC = 0.104762356, p-value = 0.6474, $p_{adj}$ = 0.9999). In line with this observation, Paneth cells in Cre+ mice remained capable of degranulation in response to the cholinergic agonist, carbachol, and secreted similar levels of lysozyme upon carbachol stimulation (*Figure 5—figure supplement 1C, D*). Thus, Paneth cells in mice lacking enteric glia exhibit morphological and functional evidence of diminished secretory function at baseline but remain competent to respond to at least some stimuli.

Baseline Paneth cell secretion in Cre+ mice could be diminished if glia are necessary for tonic Paneth cell stimulation. In the skin, another critical barrier tissue, glia are essential for the maintenance of nerve terminals, and glial depletion causes rapid and dramatic denervation (*Li and Ginty, 2014*; *Rinwa et al., 2021*). To determine if enteric glial depletion similarly causes intestinal epithelial denervation that might result in decreased Paneth cell stimulation, we characterized crypt-associated neuronal fibers in Cre+ and Cre- mice. Overall, the density of crypt innervation was no different in the two groups of mice (*Figure 5—figure supplement 1E*). Many types of intrinsic and extrinsic neurons innervate the intestinal epithelium. We found that most nerve fibers surrounding crypts are from cholinergic neurons and these fibers tightly associate with enteric glia (*Figure 5—figure supplement 1F, G*). Given that muscarinic acetylcholine receptors are among the few neurotransmitter receptors expressed by Paneth cells (*Figure 5—figure supplement 1A, B*; *Dolan et al., 2022*), cholinergic neurons are likely the ones most relevant to Paneth cell function. To test whether these neurons are

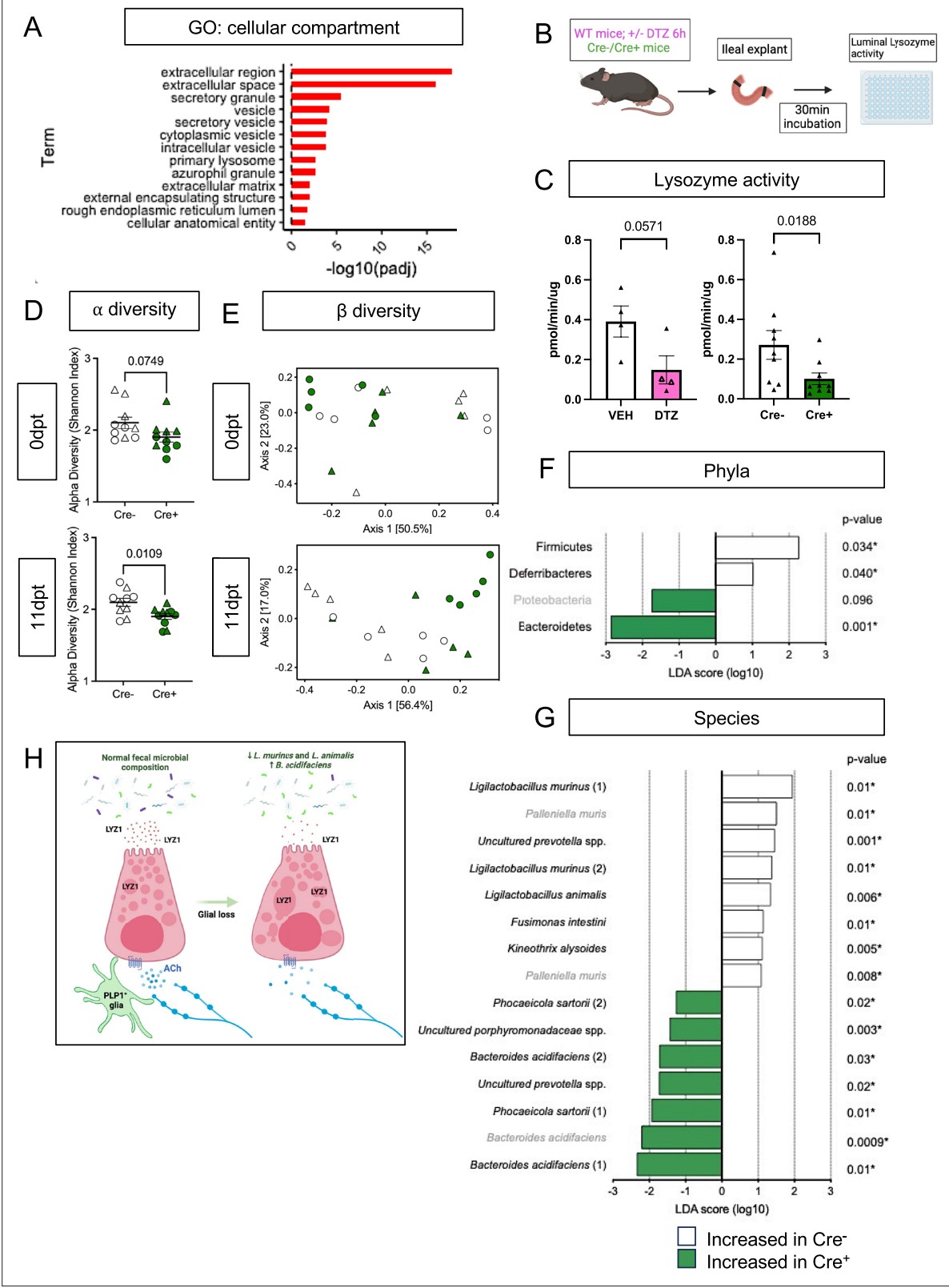

**Figure 5.** Enteric glial depletion impairs Paneth cell secretion and alters the composition of the gut microbiome. (**A**) Pathway analysis using GO term for cellular compartment shows significant enrichment of Paneth cell genes in glial-depleted mice. (**B**) Schematic of explant assay used to analyze Paneth cell secretion. Small intestinal explants were acutely isolated, ligated at both ends, and incubated in oxygenated media at 37 °C for 30 min. Luminal contents were then extracted and analyzed for lysozyme activity by fluorometric assay. Panel B created with BioRender.com. (**C**) Luminal lysozyme activity

*Figure 5 continued on next page*

*Figure 5 continued*

in ileal explants from Cre+ and Cre- mice. Lysozyme activity was lower in ileal explants from Cre+ mice compared to Cre- littermate controls (p=0.0188), mirroring the effects of Paneth cell disruption by dithizone (DTZ) in wildtype mice (p=0.0571). Each data point represents one mouse (n=4 per treatment, n=9 mice per genotype). Open triangles in DTZ group represent subset of explants incubated with 10 μM carbachol to stimulate secretion. Error bars represent SEM. ns – non-significant, p values shown are from Mann-Whitney U test. (D-G) Microbiome analysis by 16S rDNA sequencing of fecal pellets from Cre- and Cre+ mice at 0dpt (baseline, pre-induction) and 11dpt. Graphs depict α-diversity (D) and β-diversity (E) where each data point represent one mouse, with triangles indicating males and circles indicating females (n=10 for Cre- and Cre+). Error bars represent SEM. p-values reflect unpaired parametric t-. test. Analysis of phylum- (F) and species- (G) specific differences at 11dpt using LEfSe (p<0.1, LDA >1, FDR-adjusted significance values provided). Any phyla or species detected as differentially abundant at baseline are demarcated in grey. (H) Working model of glial regulation of Paneth cell function. In the normal intestine, Paneth cells are loaded with secretory granules containing LYZ1 that are released into the gut lumen in response to acetylcholine (ACh) and other signals to regulate microbial composition. Upon glial depletion, Paneth cell secretion is disrupted leading to dysmorphic granules, diminished LYZ1 secretion, and altered fecal microbial composition. This occurs without a change in Paneth cell number, loss of muscarinic acetylcholine receptor expression, or denervation of the cholinergic fibers that normally surround epithelial crypts. Panel H created with BioRender.com.

The online version of this article includes the following figure supplement(s) for figure 5:

**Figure supplement 1.** Depletion of enteric glia does not alter epithelial crypt innervation.

**Figure supplement 2.** Glial depletion alters gut microbial composition but not the spatial relationship between the host and the bacteria.

altered by glial depletion, we assessed vesicular acetylcholine transporter (VAChT) immunoreactivity in Cre[-] and Cre[+] mice. We observed no difference in the presentation of cholinergic nerve terminals that surround epithelial crypts (*Figure 5—figure supplement 1H*). These observations establish that glia are not required to maintain epithelial innervation in the intestine and that Paneth cell defects in glial-depleted mice do not result from loss of cholinergic innervation.

## Glial depletion alters gut microbiome composition

Paneth-cell-derived antimicrobial peptides are essential for preventing microbial colonization of intestinal crypts in the small intestine (*Meyer-Hoffert et al., 2008*; *Vaishnava et al., 2011*) and modulating the overall structure of the gut microbiome (*Salzman et al., 2010*; *Yu et al., 2020*). Bacterial 16S rRNA fluorescent in situ hybridization (FISH) revealed no difference in the average distance between bacteria and intestinal crypts in Cre[-] and Cre[+] mice (*Figure 5—figure supplement 2A*) indicating that glial depletion does not provoke microbial invasion into crypts. To determine if enteric glial loss alters gut microbiome composition, we performed 16S ribosomal DNA sequencing of fecal pellets from Cre[-] and Cre[+] mice at baseline (0dpt) and after glial loss (11dpt; *Figure 5—figure supplement 2B*). Both α- and β-diversity were altered by glial depletion at 11dpt in male and female mice (*Figure 5D and E*). β-diversity analysis, in particular, revealed clustering of samples by genotype at 11dpt (p=0.003), which explained a significant proportion of the inter-sample variance (R$^2$=0.25679, *Figure 5E*).

Paneth cell secretions can influence the abundance of specific members of the gut microbiome (*Salzman et al., 2010*; *Yu et al., 2020*). At the phylum level, use of linear discriminant analysis effect size (LEfSe) detected three phyla that were differentially abundant in Cre[+] mice at 11dpt but not 0dpt, with Firmicutes and Deferribacteres associated with glial presence and Bacteroidetes associated with glial depletion (*Figure 5F*; FDR-adjusted p<0.1, LDA >1). At the species level, the abundance of several taxa was altered in Cre[+] mice relative to Cre[-] controls at 11dpt (*Figure 5G*). Many of the species associated with presence of glia were *Lactobacilli* including *Ligilactobacillus murinus* and *L. animalis,* whereas species such as *Bacteroides acidifaciens* were more associated with glial ablation. Four-way analysis of the pre- and post-induction time points confirmed these changes (*Figure 5—figure supplement 2C*). *L. murinus* and *L. animalis* were previously identified among the species most depleted in the fecal microbiome of Lyz1$^{-/-}$ mice and most enriched in a Lyz1-overexpression model (*Yu et al., 2020*). Taken together, these observations indicate that genetic depletion of enteric glia disrupts Paneth cell secretion of lysozyme to impact gut microbiome composition.

## Discussion

Enteric glia secrete factors that influence intestinal epithelial cell properties in vitro, but it has remained unclear what, if any, essential roles these cells play in regulating epithelial functions in vivo. Here, we identify *PLP1*[+] cells as the glia that most closely interact with the gut epithelium and show that genetic depletion of these glia in mice does not have major effects on the intestinal transcriptome or the

cellular composition of the epithelium. Enteric glial loss, however, does cause dysregulation of Paneth cell gene expression that is associated with morphological disruption of Paneth cells, diminished lysozyme secretion, and altered gut microbial composition. Together, these observations support a working model in which glia are necessary for Paneth cell secretion of proteins that modulate the composition of the gut microbiome, but unlike in the skin, are not required for maintaining epithelial innervation (*Figure 5H*).

## Disruption of mucosal glia provokes muted and region-specific transcriptional changes in the intestine

Comparing expression of the four molecular markers used most commonly to label enteric glia, we found that SOX10 and PLP1 were the most cell-type-specific for glia in the mucosa, with little to no non-glial expression in healthy or inflamed states in both mouse and human tissues. In contrast, S100B was detectable in a subset of immune cells, while GFAP was variably expressed in the mouse mucosa and largely undetectable in human mucosal glia at the transcript and protein levels. Having identified PLP1 as the molecular marker most widely expressed by enteric glia adjacent to the epithelial layer, we utilized its promoter to probe their functional significance in adult mice using an unbiased approach. Transcriptional profiling of three different intestinal regions, quantification of cell type composition, and assessment of the histological and ultrastructural presentation of various epithelial subtypes all indicated that genetic depletion of enteric glia does not result in broad changes to the intestinal epithelium.

Our observations are contrary to some reports of the effects of *Gfap*⁺ cell depletion (*Bush et al., 1998*; *Cornet et al., 2001*; *Aubé et al., 2006*), but they are consistent with many other studies that did not uncover overt epithelial disruption when: (i) utilizing the *Plp1* or *Sox10* promoters to disrupt glia (*Rao et al., 2017*; *Yuan et al., 2020*; *Kovler et al., 2021*; *Baghdadi et al., 2022*), (ii) targeting *Gfap*⁺ cells in some cases (*Yuan et al., 2020*; *Kovler et al., 2021*), or (iii) administering chemical gliotoxins (*Aikawa and Suzuki, 1985*; *Nasser et al., 2006*). *Gfap* is often used as a marker of reactive glia in the central nervous system in the context of injury or disease. In the intestinal mucosa of human subjects with small intestinal IBD, however, *GFAP* expression appeared most robust in non-glial cells, at least at the transcriptional level (*Martin et al., 2019*). Experimental models that employ the *Gfap* promoter to disrupt enteric glia would thus presumably also affect these non-glial cells, which may explain the dramatic epithelial phenotypes reported in some previous studies. Although glia can secrete a variety of factors that modulate epithelial proliferation and barrier integrity in vitro, they do not seem essential for these functions in vivo. This may indicate the existence of redundant mechanisms to preserve these fundamental epithelial functions in vivo and/or that glial-epithelial interactions are more consequential in the context of pathophysiology than normal physiology.

The GI tract exhibits functional, cellular, and molecular specializations along its longitudinal axis. The distinct transcriptional changes resulting from the ablation of PLP1⁺ cells along this axis hint at a regional specialization of enteric glia. Consistent with this possibility, enteric glia have been shown to regulate colonic but not small intestinal GI motility (*Rao et al., 2017*), control secretomotor responses in the large intestine but not the upper GI tract (*Grubišić and Gulbransen, 2017*; *Cavin et al., 2020*), and exert different immunomodulatory roles in the small and large intestines (*Ibiza et al., 2016*; *Progatzky et al., 2021*). The region-specific functions of enteric glia as well as the mechanisms underlying this specialization will be informative to explore.

A limitation of our study and all the others to date is the lack of enteric glial-specific molecular markers and genetic promoters. All available tools to label and manipulate enteric glia also target glia in the rest of the nervous system, making it challenging to isolate their functional significance in vivo and shortening experimental timeframes. Future studies utilizing viral or intersectional genetic approaches to target enteric glia more selectively may enable a better understanding of the consequences of long-term glial disruption.

## Enteric glia as putative regulators of Paneth cells

Genetic depletion of enteric glia in adult mice provoked selective transcriptional, morphological, and ultrastructural disruption of Paneth cells, a highly secretory cell type in the small intestinal epithelium that is important for regulation of microbial ecology and innate host defense. Although the close physical association between enteric glia and small intestinal crypts in which Paneth cells reside is well

known (**Neunlist et al., 2007**; **Bush et al., 1998**; **Van Landeghem et al., 2011**), to our knowledge this is the first study linking enteric glia and Paneth cell biology. Loss of PLP1[+] enteric glia did not affect Paneth cell number but caused many of them to lose their typical morphologies and altered the appearance of their secretory granules. These morphological changes were associated with reduced luminal secretion of lysozyme, one of the most abundant AMPs produced by Paneth cells.

Morphologic changes in Paneth cells, similar to those we observed in Cre[+] mice, have been reported in studies where cholinergic signaling is blocked or vagal innervation to the intestine is severed. For example, the cholinergic antagonist atropine triggers accumulation and enlargement of Paneth cell secretory granules in mice and rats (**Satoh, 1988**; **Satoh et al., 1994**; **Sundström and Helander, 1980**). Activation or inhibition of cholinergic signaling to Paneth cells has also been shown to increase or decrease their secretory activity, respectively (**Satoh et al., 1989**; **Satoh, 1988**; **Satoh et al., 1992**; **Satoh et al., 1995**). We found that Paneth cells in glial-depleted mice remained competent to respond to cholinergic stimulation. Furthermore, unlike in the skin, glial depletion did not cause denervation. Cholinergic terminals were still present in close proximity to Paneth cells in Cre[+] mice. These observations suggest that while the infrastructure for neuroepithelial signaling remains intact in the intestines of Cre[+] mice, neurotransmission across this interface may be compromised in the absence of glia.

An alternative mechanism for glial regulation of Paneth cells is through effects on autophagy, a process important for Paneth cell secretion (**Cadwell et al., 2008**; **Wittkopf et al., 2012**; **Adolph et al., 2013**; **Bel et al., 2017**). Autophagy-related pathways were not transcriptionally enriched in glia-deficient mice, but glial-derived signals could modulate secretory autophagy in Paneth cells indirectly. For instance, Paneth cells express a receptor for IL-22, a cytokine whose production is stimulated by neurotrophic factors secreted by enteric glia (**Ibiza et al., 2016**; **Gaudino et al., 2021**). IL-22 promotes Paneth cell maturation and can license the cells for secretory autophagy in the context of *Salmonella typhimurium* infection (**Gaudino et al., 2021**). A third potential mechanism is through direct ligand-receptor interactions. Paneth cell development and function are regulated by secreted WNT proteins, which are expressed by subpopulations of enteric glia (**Baghdadi et al., 2022**). The Paneth cell phenotypes described in mouse models of disrupted WNT signaling differ from those in glial-deficient mice (**Batlle et al., 2002**; **Pinto et al., 2003**; **van Es et al., 2005**; **Ireland et al., 2004**; **Fevr et al., 2007**), but this possibility could be further explored.

Although Paneth cells are not typically found in the healthy mouse colon, differential gene expression analysis in glial-deficient mice revealed upregulation of transcripts characteristic of Paneth cells, including *Lyz1*. Although the magnitude of this upregulation was more significant than in the small intestine, the corresponding LYZ1 protein was undetectable by IHC, leaving it unclear which colonic cells upregulate *Lyz1* transcripts upon glial depletion. Colonic Paneth-like cells (PLCs) have been described in mice and humans (**Sasaki et al., 2016**; **Wang et al., 2020**). At least in humans, these PLCs can express *LYZ1* (**Wang et al., 2020**). It will be informative to assess if colonic PLCs are dysregulated by enteric glial depletion similar to Paneth cells in the small intestine.

We found that genetic depletion of enteric glia was associated with altered fecal microbiome composition within days, including reduced abundance of *L. murinus* and *L. animalis* and increased abundance of several species of *Bacteroidales*. These changes seemed to occur irrespective of the effects of glial depletion on gut motility, because they were observed in both males with normal GI transit times and females with accelerated transit. In mice engineered to either lack or overexpress LYZ1, the fecal abundance of both *L. murinus* and *L. animalis* together drops or increases, respectively (**Yu et al., 2020**). Conversely, in ZnT2-deficient mice, which exhibit reduced lysozyme activity, *Bacteroides* is the only genus significantly increased in their feces (**Podany et al., 2016**). The similarities between the shifts in microbial composition observed in these constitutive systems of Paneth cell disruption and our model of acute glial depletion support a functional link between the fecal microbial changes in Cre[+] mice and reduced LYZ1 secretion.

Overall, our results uncover a functional interaction between enteric glia and Paneth cells in the small intestine and establish a role for enteric glia in shaping gut microbial ecology. Given the strong genetic associations between Paneth cells and IBD (**Wehkamp and Stange, 2020**), and the well-established involvement of the microbiome in a wide variety of human disorders (**de Vos et al., 2022**), identifying the mechanisms underlying glial regulation of these secretory cells might reveal novel targets for tuning their activity for therapeutic benefit.

# Materials and methods

**Key resources table**

| Reagent type (species) or resource | Designation | Source or reference | Identifiers | Additional information |
|---|---|---|---|---|
| Strain, strain background (*Mus Muscularis*) | Plp1-eGFP, FVB/NJ, male and female | JAX | Cat: 033357 RRID:IMSR_JAX:033357 | |
| Strain, strain background (*M. Muscularis*) | PLP1$^{CreER}$, FVB/NJ, male and female | JAX | Cat: 005975 RRID:IMSR_JAX:005975 | |
| Strain, strain background (*M. Muscularis*) | Rosa26$^{DTA/DTA}$, C57/BL6, male and female | JAX | Cat: 009669 RRID:IMSR_JAX:009669 | |
| Strain, strain background (*M. Muscularis*) | ChAT-eGFP, C57/BL6, male and female | JAX | Cat: 007902 RRID:IMSR_JAX:007902 | |
| Chemical compound, drug | Paraformaldehyde | ThermoFisher Scientific | Cat: 28908 | Used at 4% diluted in 1 x phosphate buffered saline |
| Chemical compound, drug | UEA-I | Vector Labs | Cat: DL-1067–1 | |
| Chemical compound, drug | Vectashield | Vector Labs | Cat: H-1200 | |
| Commercial assay or kit | VECTASTAIN Elite ABC-HRP Kit PK | Vector Labs | Cat: PK-6100 | |
| Commercial assay or kit | ImmPACT DAB Substrate Kit, Peroxidase | Vector Labs | Cat: SK-4105 | |
| Antibody | Rabbit polyclonal anti CHGA | Abcam | Cat: ab-15160 RRID:AB_301704 | 1:1000 |
| Antibody | Goat polyclonal anti b-Catenin | R&D Systems | Cat: AF1329-SP RRID:AB_354736 | 1:200 |
| Antibody | Goat polyclonal anti DEFA5 | Gift from Andre Ouellette | N/A | 1:1000 |
| Antibody | Rat monoclonal anti E-cadherin | Life Tech | Cat: 13–1900 RRID:AB_2533005 | 1:400 |
| Antibody | Rabbit polyclonal anti GFAP | Sigma-Aldrich | Cat: G9269 RRID:AB_477035 | 1:500 |
| Antibody | Rabbit polyclonal anti LYZ1 | DAKO | Cat: A0099 RRID:AB_2341230 | 1:500 |
| Antibody | Rabbit polyclonal anti MUC2 | Santa Cruz Biotechnology | Cat: sc-15334 RRID:AB_2146667 | 1:200 |
| Antibody | Rat monoclonal anti PLP1/DM20 | Gift from Wendy Macklin | N/A | 1:500 |
| Antibody | Rabbit polyclonal anti S100β | DAKO | Cat: Z0311 RRID:AB_10013383 | undiluted or 1:500 |
| Antibody | Mouse monoclonal anti TUBB3 | Biolegend | Cat: 801201 RRID:AB_2313773 | 1:500 |
| Antibody | Rabbit polyclonal anti VACHT | Synaptic systems | Cat: 139 103 RRID:AB_887864 | 1:500 |
| Antibody | Rat monoclonal CD16/32 (FcR-blocking) | Biolegend | Cat: 101301, clone 93 RRID:AB_312800 | 1:50 |
| Antibody | Rat monoclonal NKM 16-2-4 | Miltenyi Biotec | Cat: 130-102-150 RRID:AB_2660295 | 1:10 |

*Continued on next page*

*Continued*

| Reagent type (species) or resource | Designation | Source or reference | Identifiers | Additional information |
|---|---|---|---|---|
| Antibody | Rat monoclonal CD326 (Ep-CAM) | Biolegend | Cat: 118213, clone G8.8 RRID:AB_1134105 | 1:50 |
| Chemical compound, drug | TRIzol | Thermofisher | Cat: 15596026 | |
| Commercial assay or kit | RNeasy Kit | Qiagen | Cat: 74004 | |
| Other | DAPI stain | Vector Laboratories | Cat: H-1200–10 | |
| Software, algorithm | R studio | R studio | RRID:SCR_000432 | |
| Software, algorithm | Deseq2 | Deseq2 | RRID:SCR_015687 | |
| Software, algorithm | DiVenn 2.0 | DiVenn 2.0, *Sun et al., 2019* | PMID:31130993 | |
| Software, algorithm | STAR | STAR | RRID:SCR_004463 | |
| Software, algorithm | featurecounts | featurecounts | RRID:SCR_012919 | |
| Software, algorithm | GSEApy | GSEApy | RRID:SCR_025803 | |
| Commercial assay or kit | iScript cDNA Synthesis Kit | BioRad | Cat: 1708890 | |
| Commercial assay or kit | SYBR Select Master Mix | Thermofisher | Cat: 4472908 | |
| Sequence-based reagent | Epcam_F | PMID:25479966 | | PCR primer; TCGCAGGTCTTCATCTTCCC |
| Sequence-based reagent | Epcam_R | PMID:25479966 | | PCR primer; GGCTGAGATAAAGGAGATGGGT |
| Sequence-based reagent | Lyz1_F | PMID:28336548 | | PCR primer; ATGGCTACCGTGGTGTCAAG |
| Sequence-based reagent | Lyz1_R | PMID:28336548 | | PCR primer; CGGTCTCCACGGTTGTAGTT |
| Software, algorithm | GraphPad Prism | GraphPad Prism | RRID:SCR_002798 | |
| software, algorithm | ImageJ | FIJI | RRID:SCR_002285 | |
| commercial assay or kit | RNAscope V2 Assay | ACDBio | Cat: 323100 | |
| sequence-based reagent | Lgr5-C1 | ACDBio | Cat: 312171 | |
| Chemical compound, drug | Opal dye 570 | Akoya Sciences | Cat: FP1488001KT | |
| Chemical compound, drug | ribonucleoside vanadyl complexes (RVC) | New England BioLabs | Cat: S1402S | |
| Chemical compound, drug | glutaraldehyde | Electron Microscopy Sciences | Cat: 111-30-8 | Used at 2% |
| Chemical compound, drug | formaldehyde | Electron Microscopy Sciences | Cat: 15700 | Used at 2.5% |
| Chemical compound, drug | cacodylate buffer | Millipore Sigma | Cat: 97068 | |
| Chemical compound, drug | uranyl acetate | Electron Microscopy Sciences | Cat: 541-09-3 | Used at 2% |
| Chemical compound, drug | lead citrate | Sigma-Aldrich | Cat: 6107-83-1 | |

*Continued on next page*

*Continued*

| Reagent type (species) or resource | Designation | Source or reference | Identifiers | Additional information |
|---|---|---|---|---|
| Chemical compound, drug | $Li_2CO_3$ | Sigma-Aldrich | Cat: 255823 | Used at 135 mM |
| Chemical compound, drug | Diphenylterazine | Sigma-Aldrich | Cat: D5130 | Used at 100 mg/kg |
| Chemical compound, drug | Carbachol | Thermo Fisher Scientific | Cat: L06674.03 | Used at 10 µM |
| Commercial assay or kit | Lysozyme Activity Assay Kit | Abcam | Cat: ab211113 | |
| Commercial assay or kit | ZymoBIOMICS – 96 DNA Kit | Zymo Research | Cat: D4309 | |
| Commercial assay or kit | DNA Clean and Concentrator TM – 5 Kit | Zymo Research | Cat: D4014 | |
| Commercial assay or kit | NEBNext Library Quant Kit | New England BioLabs | Cat: E7630 | |
| Sequence-based reagent | EUB338 | Millipore Sigma 16 s bacterial RNA probe | | GCTGCCTCCCGTAGGAGT |
| Sequence-based reagent | Nonsense control | Millipore Sigma 16 s bacterial RNA control probe | | CGACGGAGGGCATCCTCA |

## Mice

Mice were group-housed in a specific pathogen-free facility with a 12 hr dark cycle and handled per protocols approved by the Institutional Animal Care and Use Committees of Boston Children's Hospital, adherent to the NIH Guide for the Care and Use of Laboratory Animals. Drinking water and laboratory chow were provided ad libitum. Male and female littermate mice were used for most experiments except where noted (males indicated as triangles and females as circles unless stated otherwise). PLP1[CreER] mice (JAX 005975) and Plp1-eGFP mice (JAX 033357) were maintained on the FVB/NJ background while Rosa26[DTA/DTA] mice (JAX 009669) and ChAT-eGFP mice (JAX 007902) were maintained on C57BL/6 background. For generation of all experimental cohorts of glial-depleted mice, PLP1[CreER] hemizygous mice were bred with Rosa26[DTA/DTA] mice to generate PLP1[CreER] Rosa26[DTA/+] mice and Rosa26[DTA/+] littermate controls. These mice were administered 8 mg of tamoxifen in corn oil once by orogastric gavage at 5–6 weeks of age, as previously described (*Rao et al., 2017*). All analysis was carried out 11 days after tamoxifen administration (11dpt) unless indicated otherwise.

## Immunohistochemistry

For frozen sections, tissues were first fixed in 4% paraformaldehyde (PFA)/phosphate buffered saline (PBS) for 1.5 hours (hr), equilibrated in 30% sucrose/PBS and embedded, as previously described (*Rao et al., 2017*). For IHC, 10–14 µm sections of intestine were incubated in blocking solution (0.1% Triton + 5% heat-inactivated goat [HINGS] or donkey serum in PBS), incubated overnight at 4 °C in primary antibody/blocking solution, washed, and incubated for 1.5 hr at room temperature (RT) in secondary antibody or UEA-I (Vector Labs, #DL-1067–1)+DAPI. The slides were mounted in Vecta-shield (Vector Labs, #H-1200).

For IHC of small intestinal whole mounts, 2–3 cm segments of small and large intestine from Plp1-eGFP mice were dissected, washed with ice-cold PBS, fixed in 4% PFA/PBS for 1.5 hr at 4 °C, and then thoroughly washed with PBS. The samples were permeabilized with PBS, 0.5% Triton-X100, and incubated with primary antibodies in blocking buffer (5% HINGS, 20% DMSO, 0.5% PBS Triton) for 48 hr at RT. They were then washed with permeabilization solution and incubated for 24 hr with secondary antibodies +DAPI. The whole mounts were mounted in Vectashield.

For DAB immunochemistry, de-identified archived formalin-fixed paraffin-embedded female adult human small intestine tissue samples were used under the approved Beth Israel Deaconess Medical Center IRB protocol 2020P001104. The samples were sectioned and subjected to dewaxing with incubation at 58 °C for 15–20 min followed by washes in 100% xylene (2x5 min). The slides were

rehydrated in 100% ethanol bath (3x5 min) followed by 70% ethanol incubation for 10 min. Following a wash with PBS, the slides were subjected to antigen retrieval by incubation in boiling citrate buffer solution for 20 min. Subsequently, a blocking solution was applied (2.5% HINGS +2.5% BSA in 0.1% PBS-TritonX100) for 2 hr at RT. For PLP1 and GFAP staining, prior to staining, the sections were incubated with hydrogen peroxide blocking solution (Abcam, #ab64218) for 10 min at RT. Primary antibodies in the blocking solution were applied for overnight incubation at 4 °C. VECTASTAIN Elite ABC-HRP Kit PK-(Vector Labs, #PK-6100) and ImmPACT DAB Substrate Kit, Peroxidase (Vector Labs, #SK-4105) were used according to the manufacturer's instructions. Briefly, the slides were washed, incubated with biotinylated goat anti-rabbit IgG secondary antibody (1:500 in the blocking solution) for 2 hr at RT, washed, and subjected to VECTASTAIN ABC solution (prepared 30 min in advance) for 45 min at RT. Subsequently, they were washed and incubated with the DAB solution (1:30 dilution of DAB reagent in ImmPACT DAB diluent) until a visible change to brown color was observed (20 s-2min). The slides were washed and mounted in glycerol for subsequent imaging.

| Target | Supplier | Catalog number | RRID | Dilution | Fluorophore | Application |
|---|---|---|---|---|---|---|
| CHGA | Abcam | ab-15160 | RRID:AB_301704 | 1:1000 | | |
| b-Catenin | R&D Systems | AF1329-SP | RRID:AB_354736 | 1:200 | | |
| DEFA5 | Gift from A. Ouellette | N/A | N/A | 1:1000 | | |
| E-cadherin | Life Tech | 13–1900 | RRID:AB_2533005 | 1:400 | | |
| GFAP | Sigma-Aldrich | G9269 | RRID:AB_477035 | 1:500 | | |
| LYZ1 | DAKO | A0099 | RRID:AB_2341230 | 1:500 | N/A | IHC |
| MUC2 | Santa Cruz Biotechnology | sc-15334 | RRID:AB_2146667 | 1:200 | | |
| PLP1/DM20 | Gift from Wendy Macklin, Ph.D. | N/A | N/A | 1:500 | | |
| S100β | DAKO | Z0311 | RRID:AB_10013383 | undiluted or 1:500 | | |
| TUBB3 | Biolegend | 801201 | RRID:AB_2313773 | 1:500 | | |
| VACHT | Synaptic systems | 139 103 | RRID:AB_887864 | 1:500 | | |
| CD16/32 (FcR-blocking) | Biolegend | 101301, clone 93 | RRID:AB_312800 | 1:50 | N/A | |
| NKM 16-2-4 | Miltenyi Biotec | 130-102-150 | RRID:AB_2660295 | 1:10 | PE | Flow cytometry |
| CD326 (Ep-CAM) | Biolegend | 118213, clone G8.8 | RRID:AB_1134105 | 1:50 | APC | |

## RNA sequencing

All samples were collected between 9AM and 12PM. Mice were euthanized, and the GI tract was dissected into sterile, ice-cold PBS. The luminal content was flushed out of the tissue, fat and mesentery were trimmed, and then 1 cm fragments of duodenum, proximal ileum, and proximal colon were cut, immersed in TRIzol reagent (Thermofisher #15596026), homogenized, frozen on dry ice, and stored at –80 °C until RNA extraction. For the mechanical separation of ileal epithelium, 6 cm of the most distal small intestine (ileum) was used. The tissue was cut longitudinally and cleaned in ice-cold 1xPBS such that any remaining fecal/luminal content was removed. The opened ileal tissue was placed in 10 ml of 5 mM EDTA in sterile PBS, gently mixed, and incubated on ice in a horizontal position for 10 min while ensuring its complete submersion in the EDTA solution. Halfway through, the tube was gently tilted twice to mix. Subsequently, the EDTA solution was decanted, and the tissue was washed with 10 ml of sterile HBSS twice. To mechanically separate the epithelial fraction, the tissues were extended epithelium-side-up on a glass slide and the epithelial layers (villi first, followed by crypts) were separated using a bent 20 G needle. The epithelial content was immediately transferred to the TRIzol reagent, homogenized, frozen on dry ice, and stored at –80 °C until RNA extraction. RNA

was extracted using phenol/chloroform extraction methods followed by a cleanup with the RNeasy Kit (QIAGEN #74004). RNA samples were analyzed for purity and concentration and submitted to Novogene Corporation Inc (Sacramento, CA, United States) for quality control, library construction, and sequencing. Sequencing was performed on Novaseq 6000 platform (20 M/PE150).

## RNA-seq analysis

We used trimmomatic (*Bolger et al., 2014*) to trim the low-quality next generation sequencing (NGS) reads (-threads 20 ILLUMINACLIP:TruSeq3-PE.fa:2:30:10 LEADING:3 TRAILING:3 SLIDINGWINDOW: 4:15 MINLEN:36). Subsequently, only the high-quality trimmed reads were aligned to the mouse reference genome using STAR (*Dobin et al., 2013*). The reads counts were calculated by feature-Counts software (*Liao et al., 2014*). Differentially expressed genes (DEGs) were identified by using the DESeq2 R package (adjusted p-value $\leq 0.05$; *Love et al., 2014*). For analysis of shared gene expression, DiVenn analysis was carried out as previously described (*Sun et al., 2019*). GSEA analysis was performed using GSEApy (*Fang et al., 2023*).

## Quantitative PCR

For validation of RNAseq results, two separate cohorts of mice were used. The tissues were dissected and processed as described above. RNA was extracted using phenol/chloroform extraction methods followed by a cleanup with the RNeasy Kit (QIAGEN #74004). The RNA was converted to cDNA using iScript cDNA Synthesis Kit (Bio-Rad #1708890) and the qPCR was run with SYBR Select Master Mix (Thermo Fisher # 4472908). The following primers were used.

| Target | Forward primer | Reverse primer | Annealing temp. |
|---|---|---|---|
| Epcam | TCGCAGGTCTTCATCTTCCC | GGCTGAGATAAAGGAGATGGGT | 60 °C |
| Lyz1 | ATGGCTACCGTGGTGTCAAG | CGGTCTCCACGGTTGTAGTT | 58 °C |

## Imaging and cell quantification

Image acquisition was carried out by investigators blinded to genotype. Animals of both sexes were analyzed. Data was analyzed using Microsoft Excel and the GraphPad Prism program (GraphPad Software, Inc). For quantification of Alcian Blue[+] goblet cells, images were obtained from Cre[+] (n=8) and Cre[-] (n=10) animals. The number of Alcian Blue[+] cells per villus-crypt unit was counted and averaged per mouse. For LYZ1[+] Paneth cells, images of at least 50 crypts in the ileum were obtained for each Cre[+] (n=4) and Cre[-] (n=5) animals. The number of LYZ1[+] cells per crypt was counted and averaged per mouse. For Chromogranin A[+] EECs, images of at least 42 villi and 100 crypts in the ileum were obtained, for each Cre[+] (n=3 per group) and Cre[-] (n=4 per group) animal. The number of CHGA[+] cells per villus-crypt unit was counted and averaged per mouse. For Lgr5[+] cells, 6–9 z-stack images (20X) were obtained for each Cre[+] (n=4) and Cre[-] (n=4) animal. For quantification of mean fluorescence intensity (MFI), z-stacks were subjected to maximum intensity projection, and ROI's were drawn from the villus base to crypt base that defined crypt regions. MFI of the ROI was calculated and averaged per mouse. For quantification of crypt innervation, 15 random z-stack images of individual crypts were obtained for each Cre[+] (n=4) and Cre[-] (n=4) animal. Z-stack images were subjected to maximum intensity projection, binarization, thresholding, and smoothening. Subsequently, the signal was converted to masks, and the percentage area of TUBB3 signal coverage of the image field was calculated and averaged per mouse. A Zeiss LSM 880 confocal microscope was used to acquire images for all fluorescent IHC except for CHGA[+] EECs, for which a Leica DM6000B epifluorescent microscope was used.

## Flow cytometry

Flow cytometry of M cells from Peyer's Patches (PPs) was adapted from *Gicheva et al., 2016* Briefly, the entire length of jejunum and ileum were dissected from Cre[-] and Cre[+] mice. Then, 6 Peyer's patches were harvested per mouse, placed in 1.5 mL microcentrifuge tubes with ice-cold 1xPBS, and vortexed vigorously to remove debris. After three PBS washes, the PPs were placed in 10 ml of PBS with 5 mM EDTA and 1 mM DTT for 30 min at 37 °C. The samples were additionally triturated to aid the dissociation. Following digestion, the cell suspension was filtered through a 40 µm strainer, centrifuged at 475 × *g* for 5 min, and incubated with FcR-blocking antibody on ice for 10 min. The

cells were then stained to label M-cells (NKM 16-2-4) and epithelial cells (EpCAM) in FACS buffer (2% FBS +1 mM EDTA) for 30 min at 4 °C. Subsequently, the cells were washed in FACS buffer and stained with DAPI (0.3 μg/mL). The proportion of PE$^+$ APC$^+$ DAPI$^-$ cells out of APC$^+$ DAPI$^-$ cells was determined on BD LSRFortessa.

## RNAscope

Intestinal tissue was dissected into ice-cold 1 x PBS + 4 mM ribonucleoside vanadyl complexes (RVC) to inhibit RNAse activity and flushed to remove fecal content. Segments of ~2 cm were fixed in RNAse-free 4%PFA/PBS for 24 hr and incubated overnight in 30% sucrose. The tissue was embedded in pre-chilled OCT and frozen on dry ice. The tissues were stored at –80 °C. For staining, 8 μm slices were sectioned and air dried at –20 °C. RNAscope V2 (ACDBio #323100) was used for in situ hybridization. Briefly, the slides were washed in 1xPBS, incubated for 30 min at 60 °C, and post-fixed with pre-chilled 4%PFA for 15 min at 4 °C. Subsequently, the slides were dehydrated in increasing concentrations of RNase-free EtOH (50%, 70%, and twice 100% for 5 min each), treated with hydrogen peroxide (RNAscope Hydrogen Peroxide Reagent) for 10 min at RT, and washed in DEPC-treated water. For antigen retrieval, the slides were immersed in boiling hot RNAscope Target Retrieval Reagent, and incubated at 99 °C for 5 min. Following a wash in RT DEPC-treated water, they were incubated in 100% EtOH for 3 min, dried at RT, and subjected to protease treatment (RNAscope Protease III Reagent) for 30 min at 40 °C. After two washes with 1 x PBS, probe hybridization and signal amplification were carried out according to RNAscope Multiplex Fluorescent V2 Assay using Lgr5-C1 probe and Opal dye 570 (Akoya Sciences). Slides were mounted with Vectashield and DAPI.

## Electron microscopy

Tissues were excised, washed in PBS, cut along the mesenteric plane, pinned flat, and then fixed in 2% glutaraldehyde (Electron Microscopy Sciences, Hatfield PA) and 2.5% formaldehyde (Electron Microscopy Sciences) in 0.1 M cacodylate buffer pH 7.4 containing 0.1 mM EGTA for 10 min at RT with gentle flushing. The tissue was then cut into small pieces, and fixed for an additional 1 hr in the same fixative at RT. Tissues were washed with 0.1 M cacodylate buffer, and then loaded into a planchette (Technotrade International, Manchester, NH) with PBS containing 20% BSA and 5% FBS, and subjected to high-pressure freezing using a Wohlwend High Pressure Freezer (Technotrade International). Rapid freeze substitution, as described (*McDonald, 2014*), was done using 1% osmium tetroxide, 0.5% uranyl acetate, 95% acetone and 5% dH2O. After freeze substitution, the tissue was infiltrated with graded acetone into LX112 resin (Ted Pella, Inc Redding, CA). Ultrathin sections were cut with a Leica Ultracut E ultramicrotome (Leica Microsystems, Wetzlar Germany), placed on formvar and carbon coated grids, and then stained with 2% uranyl acetate (Electron Microscopy Sciences) and lead citrate (Sigma-Aldrich). Grids from each treatment were imaged using a JEOL 1400 electron microscope (JEOL USA, Peabody, MA) equipped with an Orius SC1000 digital CCD camera (Gatan, Pleasanton, CA).

## Paneth cell secretion assay

All Paneth cell secretion assays were carried out from 9AM to 12PM with four mice per assay except for when DTZ was administered. For DTZ experiments, the mice were administered vehicle (135 mM Li$_2$CO$_3$ solution; Sigma, #255823) or 100 mg/kg DTZ (Sigma, # D5130) diluted in vehicle six hours before the start of the explant experiment. Mice were euthanized and 10 cm of distal small intestine was dissected into sterile, ice-cold PBS. The luminal content was flushed out of the tissue, and fat and mesentery were trimmed. A 6.5–7 cm fragment of the most distal small intestine was isolated and the remaining PBS was removed from the tissue. One end of the tissue was firmly tied and 150 μl of sterile, ice-cold PBS was pipetted into the intestinal tube. The open end of the intestine was firmly tied to create a closed cylinder filled with PBS. The length of the tissue, from one tied end to the other, was measured. The process was repeated for all samples which were kept in sterile, ice-cold PBS. The explants were subsequently placed in oxygenated Krebs at 37 °C and incubated continuously bubbled with Carbogen for the duration of the experiment (30 min). For experiments involving carbachol, the compound was added at a concentration of 10 μM at the beginning of incubation. Following incubation, one at a time, the tissues were opened, and the luminal contents were extracted. The volume of recovered solution was measured and diluted in sterile PBS as necessary to get to a final volume

of 25 µL/cm of intestine. The samples were sterile-filtered with pre-wetted 0.22 µm syringe filters. Lysozyme activity was measured using the Lysozyme Activity Assay Kit (Abcam #ab211113) according to the manufacturer's instructions.

## 16S ribosomal DNA (rDNA) gene phylotyping

Male and female PLP1[CreER] Rosa26[DTA/+] and Rosa26[DTA/+] littermate mice were group-housed segregated by sex and genotype from the time of weaning. Two to four spontaneously expelled fecal pellets were collected from each mouse at 9-10AM at two timepoints: 0dpt (prior to tamoxifen administration) and 11dpt. Fecal samples were immediately frozen and stored at –80 °C. Genomic DNA for 16 S rDNA amplicon next generational sequencing was isolated using the ZymoBIOMICS – 96 DNA Kit (Zymo Research, D4309). The 16S amplicon library was prepared in a 96-well format using dual-index barcodes (*Rao et al., 2021*). Libraries were cleaned with the DNA Clean and Concentrator TM – 5 Kit (Zymo Research, D4014) and then quantified by qPCR (NEBNext Library Quant Kit, NEB, E7630). 20 pM of DNA were loaded onto an Illumina MiSeq (v3, 600 cycle) and sequenced. To generate the Operational Taxonomic Unit (OTU) table for analyses of gut microbiome composition and diversity, Illumina raw reads were de-multiplexed, paired end joined, adapter trimmed, quality filtered, dereplicated, and denoised. Sequences were mapped against the publicly available 16 S rDNA databases SILVA and UNITE and clustered into OTUs ≥ 97% nucleotide sequence identity. OTU-based microbial community diversity was estimated by calculating Shannon's alpha diversity index and Bray-Curtis beta diversity index. Differential abundance analyses were performed with LEfSe with significantly different features having an alpha value less than or equal to 0.1 and a logarithmic LDA score greater than or equal to 1. Stratifying the data by sex within-sample revealed no major sex-specific differences in microbiome diversity or enriched/depleted biomarkers in the core genotype-dependent observations.

## 16S bacterial rRNA FISH

16 S rRNA FISH was carried out as described previously with some modifications (*McGuckin and Thornton, 2012*). Briefly, distal small intestine was dissected from Cre[-] and Cre[+] mice directly into methanol-Carnoy's fixative [60% (v/v) dry methanol, 30% (v/v) chloroform, 10% (v/v) glacial acetic acid]. Care was taken to limit exposure to aqueous solutions. The samples were kept in the fixative solution for 72 hr followed by washes with 100% methanol (2x30 min), 100% ethanol (3x30 minutes), xylene (2x20–30 min), paraffin (2x20–30 min). The samples were embedded in paraffin, sectioned, and stained as previously described (*Johansson and Hansson, 2012*). Briefly, the sections were dewaxed by incubating at 60 °C for 10 min, followed by two xylene baths (1x10 min at 60 °C, 1x10 min at RT). The sections were incubated in 99.5% ethanol for 5 min, air dried, and stained with EUB338 or a control probe in a hybridization solution at 50 °C overnight. Subsequently, the sections were washed and subjected to immunohistochemistry protocol as described below.

|  | Fluorophore | Sequence |
|---|---|---|
| EUB338 | Cy3 | GCTGCCTCCCGTAGGAGT |
| Nonsense control | Cy3 | CGACGGAGGGCATCCTCA |

## Experimental design and statistical analysis

Both R 4.2.0 and Prism were used for statistical analyses and graphical visualization. All experiments were performed blinded to experimental conditions during data collection and analysis. Data points represent biological replicates, with each replicate obtained from a different animal. Unless stated otherwise, data were collected from a single experiment. Sample sizes were determined based on previous studies from our group or established based on preliminary observations. For pairwise comparisons, an unpaired parametric t-test or Mann-Whitney U test was used after testing for equal variance between the groups unless stated otherwise. If variance was significantly different, unpaired parametric t-test with Welsh Correction was applied. For comparisons between more than two groups, one-way ANOVA with Tukey multiple comparisons test was used.

## Acknowledgements

We are grateful to the funding sources listed below, Wendy B Macklin (University of Colorado) for PLP-1/DM20 antibody, Andre J Ouellette (University of California, Irvine) for the DEFA5 antibody, and members of the Rao laboratory for discussions and experimental support. We thank Michael Grey and Michael Rutlin for critical reading of the manuscript. The RNA-seq analysis was performed with the computational resources provided by the Research Computing Group at Boston Children's Hospital and Harvard Medical School (Boston, MA), including High-Performance Computing Clusters Enkefalos 2 (E2), and the BioGrids scientific software made available for data analysis. This study was supported by the Schmidt Science fellowship (AS), NSF graduate fellowship (AM), Smith Family Foundation Odyssey Award (MR), NDSEG fellowship (GAK), and NIH R01DK130836, K08DK110532, and R01DK135707 (MR). Core facilities utilized were supported by the Harvard Digestive Disease Center (NIH P30DK034854).

## Additional information

### Funding

| Funder | Grant reference number | Author |
| --- | --- | --- |
| National Institutes of Health | R01DK130836 | Meenakshi Rao |
| National Institutes of Health | K08DK110532 | Meenakshi Rao |
| National Institutes of Health | R01DK135707 | Meenakshi Rao |
| Schmidt Family Foundation | Fellowship | Amy Shepherd |
| National Science Foundation | Graduate Research Fellowship Program | Anoohya N Muppirala |
| National Defense Science and Engineering Graduate | Fellowship | Gavin A Kuziel |
| Smith Family Foundation | Odyssey Award | Meenakshi Rao |

The funders had no role in study design, data collection and interpretation, or the decision to submit the work for publication.

### Author contributions

Aleksandra Prochera, Conceptualization, Investigation, Writing – original draft; Anoohya N Muppirala, Conceptualization, Investigation, Writing – review and editing, Performed RNA-sequencing analysis; Gavin A Kuziel, Conceptualization, Investigation, Writing – review and editing, Performed 16S microbiome analysis; Salima Soualhi, Amy Shepherd, Conceptualization, Investigation, Writing – review and editing; Liang Sun, Formal analysis, Writing – review and editing, Performed RNA-sequencing analysis; Biju Issac, Formal analysis, Performed RNA-sequencing analysis; Harry J Rosenberg, Farah Karim, Kristina Perez, Kyle H Smith, Investigation, Writing – review and editing; Tonora H Archibald, Investigation; Seth Rakoff-Nahoum, Supervision, Writing – review and editing; Susan J Hagen, Supervision, Investigation, Methodology, Writing – review and editing; Meenakshi Rao, Conceptualization, Supervision, Funding acquisition, Investigation, Writing – original draft, Writing – review and editing

### Author ORCIDs

Aleksandra Prochera ⦿ https://orcid.org/0009-0006-5203-9507
Anoohya N Muppirala ⦿ https://orcid.org/0000-0002-6810-4127
Biju Issac ⦿ https://orcid.org/0000-0002-7311-678X
Meenakshi Rao ⦿ https://orcid.org/0000-0002-8194-9204

### Ethics

This study was performed in strict accordance with the recommendations in the Guide for the Care and Use of Laboratory Animals of the National Institutes of Health. All of the animals were handled

according to approved institutional animal care and use committee (IACUC) protocols of Boston Children's Hospital (#18-12-3841).

Reviewer #1 (Public review): https://doi.org/10.7554/eLife.97144.3.sa1
Reviewer #2 (Public review): https://doi.org/10.7554/eLife.97144.3.sa2
Author response https://doi.org/10.7554/eLife.97144.3.sa3

## Additional files

### Supplementary files
MDAR checklist

Supplementary file 1. Cell type signatures derived from sc-RNAseq of intestinal epithelial cells used in *Figure 3B* and *Figure 3—figure supplement 1D*.

Supplementary file 2. Cell type signatures derived from bulk-RNAseq profiling of individual cell types (purified by flow sorting) used in *Figure 3—figure supplement 1E–F*.

Source data 1. Source data containing raw values and statistical analyses used for image quantification and qPCR analysis.

### Data availability
RNA sequencing data from glial-depleted mice are deposited in the Gene Expression Omnibus (GEO: GSE280442) and 16S bacterial rRNA datasets have been deposited at the National Center for Biotechnology Information Sequence Read Archive (BioProject Accession: PRJNA1234316). The bulk and single-cell RNA sequencing data sets analyzed from previously published studies and accession numbers are listed in the figure legends, supplementary files, or methods. All other data are available in the manuscript and the supplementary files.

The following datasets were generated:

| Author(s) | Year | Dataset title | Dataset URL | Database and Identifier |
|---|---|---|---|---|
| Prochera A, Muppirala AN, Kuziel GA, Soualhi S, Shepherd A, Sun L, Issac B, Rosenberg HJ, Karim F, Perez K, Smith KH, Archibald TH, Rakoff-Nahoum S, Hagen SJ, Rao M | 2024 | Enteric glia regulate Paneth cell secretion and intestinal microbial ecology | https://www.ncbi.nlm.nih.gov/geo/query/acc.cgi?acc=GSE280442 | NCBI Gene Expression Omnibus, GSE280442 |
| Prochera A, Muppirala AN, Kuziel GA, Soualhi S, Shepherd A, Sun L, Issac B, Rosenberg HJ, Karim F, Perez K, Smith KH, Archibald TH, Rakoff-Nahoum S, Hagen SJ, Rao M | 2025 | Enteric glia regulate Paneth cell secretion and intestinal microbial ecology | https://www.ncbi.nlm.nih.gov/bioproject/?term=PRJNA1234316 | NCBI BioProject, PRJNA1234316 |

The following previously published datasets were used:

| Author(s) | Year | Dataset title | Dataset URL | Database and Identifier |
|---|---|---|---|---|
| Lee HO, Hong Y, Etlioglu HE, Cho YB, Pomella V, Van den Bosch B, Vanhecke J, Verbandt S, Hong H, Min JW, Kim N | 2020 | Single cell 3' RNA sequencing of 23 Korean colorectal cancer patients | https://www.ncbi.nlm.nih.gov/geo/query/acc.cgi?acc=GSE132465 | NCBI Gene Expression Omnibus, GSE132465 |

*Continued*

| Author(s) | Year | Dataset title | Dataset URL | Database and Identifier |
|---|---|---|---|---|
| Nyström EE, Martinez-Abad B, Arike L, Birchenough GM, Nonnecke EB, Castillo PA, Svensson F, Bevins CL, Hansson GC, Johansson ME | 2021 | Gene expression profile of goblet cells (GCs) from the distal colon (DC) and the 8th portion of the small intestine (Si8) | https://www.ncbi.nlm.nih.gov/geo/query/acc.cgi?acc=GSE144363 | NCBI Gene Expression Omnibus, GSE144363 |
| Yan KS, Gevaert O, Zheng GX, Anchang B, Probert CS, Larkin KA, Davies PS, Cheng ZF, Kaddis JS, Han A, Roelf K | 2017 | Bulk cell RNAseq of putatative intestinal stem cell populations | https://www.ncbi.nlm.nih.gov/geo/query/acc.cgi?acc=GSE99815 | NCBI Gene Expression Omnibus, GSE99815 |
| Kimura S, Nakamura Y, Kobayashi N, Shiroguchi K, Kawakami E, Mutoh M, Takahashi-Iwanaga H, Yamada T, Hisamoto M, Nakamura M, Udagawa N | 2019 | Gene expression profiling of GP2+ M cells and other epithelial cells in Peyer's patch | https://www.ncbi.nlm.nih.gov/geo/query/acc.cgi?acc=GSE108529 | NCBI Gene Expression Omnibus, GSE108529 |
| Zheng HB, Doran BA, Kimler K, Yu A, Tkachev V, Niederlov V, Cribbin K, Fleming R, Bratrude B, Betz K, Cagnin L | 2021 | PREDICT 2021 paper: FGID | https://singlecell.broadinstitute.org/single_cell/study/SCP1422/predict-2021-paper-fgid?genes=DOK1&cluster=tSNE_Coordinates_PREDICT_3p_FGID.csv&spatialGroups=--&annotation=cell_type__ontology_label--group--study&subsample=all#study-visualize | Single Cell Portal, SCP1422/predict-2021-paper-fgid |
| Martin JC, Chang C, Boschetti G, Ungaro R, Giri M, Grout JA, Gettler K, Chuang LS, Nayar S, Greenstein AJ, Dubinsky M | 2019 | Single-cell analysis of Crohn's disease lesions identifies a pathogenic cellular module associated with resistance to anti-TNF therapy | https://www.ncbi.nlm.nih.gov/geo/query/acc.cgi?acc=GSE134809 | NCBI Gene Expression Omnibus, GSE134809 |
| Roulis M, Kaklamanos A, Schernthanner M, Bielecki P, Zhao J, Kaffe E, Frommelt LS, Qu R, Knapp MS, Henriques A, Chalkidi N | 2019 | Paracrine orchestration of intestinal tumorigenesis by a confined mesenchymal niche | https://www.ncbi.nlm.nih.gov/geo/query/acc.cgi?acc=GSE142431 | NCBI Gene Expression Omnibus, GSE142431 |
| Haber AL, Biton M, Rogel N, Herbst RH, Shekhar K, Smillie C, Burgin G, Delorey TM, Howitt MR, Katz Y, Tirosh | 2017 | A single-cell survey of the small intestinal epithelium | https://www.ncbi.nlm.nih.gov/geo/query/acc.cgi?acc=GSE92332 | NCBI Gene Expression Omnibus, GSE92332 |
| Yu S, Tong K, Balasubramanian I, Yap GS, Ferraris RP, Bonder EM, Verzi MP, Gao N | 2018 | Paneth cells acquire multi-potency upon Notch activation after irradiation | https://www.ncbi.nlm.nih.gov/geo/query/acc.cgi?acc=GSE113536 | NCBI Gene Expression Omnibus, GSE113536 |

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
