## [Editor Report · eLife Assessment]

This study presents **important** findings on the function of enteric glia expressing proteolipid protein 1 (PLP1+ glia). The evidence supporting the claims of the authors is **solid**, although the inclusion of additional data showing the mechanisms by which PLP1+ enteric glia acts on Paneth cells would have strengthened the study. The work will be of interest to colleagues studying intestinal biology.

---

## [Referee Report · Reviewer #1 (Public review)]

The role of enteric glial cells in regulating intestinal mucosal functions at steady state has been a matter of debate in recent years. Enteric glial cell heterogeneity and related methodological differences likely underlie the contrasting findings obtained by different laboratories. Here, Prochera and colleagues used Plp1-CreERT2 driver mice to deplete the vast majority of enteric glia from the gut, and performed an elegant set of transcriptomic, microscopic and biochemical essays to examine the impact of enteric glia loss. It was found that enteric glia depletion has very limited effects on the transcriptome of gut cells 11 days after tamoxifen treatment (used to induce Diphtheria Toxin A expression in the majority of enteric glia including those present in the mucosa), and by extension - more specifically, has only minimal impact on cells of the intestinal mucosa. Interestingly, in the colon (where Paneth cells are not present) they did observe transcriptomic changes related to Paneth cell biology. Although no overt gene expression alterations were found in the small intestine - also not in Paneth cells - morphological, ultrastructural and functional changes were detected in the Paneth cells of enteric glia-depleted mice. In addition, and likely related to impaired Paneth cell secretory activity, enteric glia-depleted mice also show alterations in intestinal microbiota composition. This is an excellent study that convincingly demonstrates a role for enteric glia in supporting Paneth cells of the intestinal mucosa, suggesting that enteric glial cells shape host-microbiome interactions via the regulation of Paneth cell homeostasis.

---

## [Referee Report · Reviewer #2 (Public review)]

This is an excellent and timely study from the Rao lab investigating the interactions of enteric glia with the intestinal epithelium. Two early studies in the late 90's and early 2000's had previously suggested that enteric glia play a pivotal role in control of the intestinal epithelial barrier, as their ablation using mouse models resulted in severe and fatal intestinal inflammation. However, it was later identified that these inflammatory effects could have been an indirect product of the transgenic mouse models used, rather than due to the depletion of enteric glia. In previous studies from this lab, the authors had identified expression of PLP1 in enteric glia, and its use in CRE driver lines to label and ablate enteric glia.

In the current paper, the authors carefully examine the role of enteric glia by first identifying that PLP1-creERT2 is the most useful driver to direct enteric glial ablation, in terms of the quantity of glial cells targeted, their proximity to the intestinal epithelium, and the relevance for human studies (GFAP expression is rather limited in human samples in comparison). They examined gene expression changes in different regions of the intestine using bulk RNA-seq following ablation of enteric glia by driving expression of diptheria toxin A (PLP1-creERT2;Rosa26-DTA). Alterations in gene expression were observed in different regions of the gut, with specific effects in different regions. Interestingly, while there were gene expression changes in the epithelium, there were limited changes to the proportions of different epithelial cell types identified using immunohistochemistry in control vs glial-ablated mice. The authors then focused on investigation of Paneth cells in the ileum, identifying changes in the ultrastructural morphology and lysozyme activity. In addition, they identified alterations in gut microbiome diversity. As Paneth cells secrete antimicrobial peptides, the authors conclude that the changes in gut microbiome are due to enteric glia-mediated impacts on Paneth cell activity.

Overall, the study is excellent and delves into the different possible mechanisms of action, including investigation of changes in enteric cholinergic neurons innervating the intestinal crypts. The use of different CRE-drivers to target enteric glial cells has led to varying results in the past, and the authors should be commended on how they address this in the Discussion.

Comments on the latest version:

Thanks to the authors for addressing my concerns. The additional stratification of male vs female microbiome data was very helpful.

---

## [Author Response]

The following is the authors’ response to the original reviews.

**Reviewer #1 (Public Review):**
The role of enteric glial cells in regulating intestinal mucosal functions at a steady state has been a matter of debate in recent years. Enteric glial cell heterogeneity and related methodological differences likely underlie the contrasting findings obtained by different laboratories. Here, Prochera and colleagues used Plp1-CreERT2 driver mice to deplete the majority of enteric glia from the gut. They found that glial loss has very limited effects on the transcriptome of gut cells 11 days after tamoxifen treatment (used to induce DTA expression), and by extension - more specifically, has only minimal impact on cells of the intestinal mucosa. Interestingly, in the colon (where Paneth cells are not present) they did observe transcriptomic changes related to Paneth cell biology. Although no overt gene expression alterations were found in the small intestine - also not in Paneth cells - morphological, ultrastructural, and functional changes were detected in the Paneth cells of enteric glia-depleted mice. In addition, and possibly related to Paneth cell dysfunction, enteric glia-depleted mice also show alterations in intestinal microbiota composition.In their analyses of enteric glia from existing single-cell transcriptomic data sets, it is stated that these come from 'non-diseased' humans. However, the data on the small intestine is obtained from children with functional gastrointestinal disorders (Zheng 2023). Data on colonic enteric glia was obtained from colorectal cancer patients (Lee 2020). Although here the cells were isolated from non-malignant regions, saying that the large intestines of these patients are nondiseased is probably an overstatement.

In the Zheng et al. dataset, “functional GI disorders” refers to biopsies from children that do not have any histopathologic evidence of digestive disease. The children do, however, have at least one GI symptom that prompted a diagnostic endoscopy with biopsies, leading to the designation of “functional” disorder. Given that diagnostic endoscopies are invasive procedures that necessitate anesthesia, obtaining biopsies from asymptomatic children without any clinical indication would not be allowable per most institutional review boards, leading the authors of that study to use these samples as a control group. We had thus used the “non-diseased” label to encompass these samples as well as those from the unaffected regions of large intestine from colorectal cancer patients. We now recognize, however, that this label could be misleading, so we have revised the Results and Figure Legends to more accurately reflect details of control tissue origin for this and the Lee *et al*. (2020) datasets. Per the reviewer’s suggestion, we have removed the term “non-diseased”.

Another existing dataset including human mucosal enteric glia of healthy subjects is presented in Smillie et al (2019). It would be interesting to see how the current findings relate to the data from Smillie et al.

Per the reviewer’s suggestion, we have now added an analysis of the Smillie *et al*. dataset in Supp. Fig. 1B. This dataset derives from colonic mucosal biopsies from 12 healthy adults (8480 stromal cells) and 18 adults with ulcerative colitis (10,245 stromal cells from inflamed bowel segments and 13,147 from uninflamed), all between the ages of 20-77 years. These data show that *SOX10, PLP1*, and *S100B* are selectively expressed within the putative glial cluster from colonic mucosa of both healthy adults and individuals with ulcerative colitis, whereas GFAP is not detected (Supp. Fig. 1B). These observations are consistent with our observations from the two other human datasets already included in our manuscript in Fig. 1 and Supp. Fig. 1.

The time between enteric glia depletion and analyses (mouse sacrifice) must be a crucial determinant of the type of effects, and the timing thereof. In the current study 11 days after tamoxifen treatment was chosen as the time point for analyses, which is consistent with earlier work by the lab using the same model (Rao et al 2017). What would happen when they wait longer than 11 days after tamoxifen treatment? Data, not necessarily for all parameters, on later time points would strengthen the manuscript significantly.

This is an excellent question, particularly given the longer-lived nature of Paneth cells relative to other epithelial cell types. As detailed in our previous study, Cre^+^ mice in the Plp1CreER-DTA model are well-appearing and indistinguishable from their Cre-negative control littermates through 11dpt. Unfortunately, a limitation of the model is that beyond 11dpt, Cre^+^ mice become anorexic, lose body weight, and have signs of neurologic debility such as hindlimb weakness and uncoordinated gait. These deficits are overt by 14dpt and likely due to targeting Plp1^+^ glia outside the gut, such as Schwann cells and oligodendrocytes (as described in another study which used a similar model to study demyelination in the central nervous system, PMID: 20851998). Given these CNS effects and that starvation is well known to affect Paneth cell phenotypes (PMIDs: 1167179, 21986443), we elected not to examine timepoints beyond 11dpt. Technological advances that enable more selective cell depletion will allow study of chronic effects of enteric glial loss in the future.

The authors found transcriptional dysregulation related to Paneth cell biology in the colon, where Paneth cells are normally not present. Given the bulk RNA sequencing approach, the cellular identity in which this shift is taking place cannot be determined. However, it would be useful if the authors could speculate on which colonic cell type they reckon this is happening in.

Per the reviewer’s suggestion, we have added a paragraph to the Discussion addressing one plausible hypothesis to explain this observation. Paneth-like cells have been described in the large intestine and are known, particularly in humans, to express markers typical of Paneth cells, such as lysozyme and defensins (PMID: 27573849, 31753849). These cells could represent the source of the Paneth cell-like transcriptional signature observed in our model. Alternatively, ectopic expression of Paneth cell-associated genes in the colon has been documented in certain pathological conditions, such as colorectal cancer models (e.g., PMID: 15059925), where changes in the local microenvironment appear to trigger activation of Paneth cell genes. Similar, yet unidentified changes in our model could potentially underlie the transcriptional dysregulation related to Paneth cell biology observed here.

On the other hand, enteric glia depletion was found to affect Paneth cells structurally and functionally in the small intestine, where transcriptional changes were initially not identified. Only when performing GSEA with the in silico help of cell type-specific gene profiles, differences in Paneth cell transcriptional programs in the small intestine were uncovered. A comment on this discrepancy would be helpful, especially for the non-bioinformatician readers among us.

Standard differential gene expression analysis (DEG) of the effects of glial loss revealed significant differences only in the colon, and even then, only a handful of genes were changed. These changes were not accompanied by corresponding changes at the protein level, at least as detectable by IHC. In the small intestine, there were no significant differences by standard DEG thresholds. Unlike DEG, gene set enrichment analyses (GSEA), provides a significance value based on whether there is a higher than chance number of genes that are changing in a uniform direction without consideration for the significance of the magnitude of change. Therefore, the GSEA detected that a significant number of genes in the curated Paneth cell gene list exhibited a positive fold change difference in the bulk RNA sequencing data. This prompted us to examine Paneth cells and other epithelial cell types in more detail by IHC, functional and ultrastructural analyses, which all converged on the observation that Paneth cells were relatively selectively disrupted in the epithelium of glial depleted mice.

From looking at Figure 3B it is clear that Paneth cells are not the only epithelial cell type affected (after less stringent in silico analyses) by enteric glial cell depletion. Although the authors show that this does not translate into ultrastructural or numerical changes of most of these cell types, this makes one wonder how specific the enteric glia - Paneth cell link is. Besides possible indirect crosstalk (via neurons), it is not clear if enteric glia more closely associate with Paneth cells as compared to these other cell types. Immunofluorescence stainings of some of these cells in the Plp1-GFP mice would be informative here.

Enteric glia have long been reported to closely associate with crypts, the sites of residence for Paneth cells and intestinal stem cells (PMID: 7043279, 16423922). Consistent with these reports, our observations from Plp1-eGFP mice confirm that enteric glia often appose the entire base of small intestinal crypts (see Author response image 1 below). Given this reproducible observation, we did not pursue histological quantification to compare preferential glial apposition to specific epithelial cell types. Enteric glia have been reported to form close associations with enteroendocrine cells as well (PMID: 24587096), which is not surprising because these cells are highly innervated; however, our analyses did not reveal changes in the abundance and morphology of these cells or other epithelial cell types.

**Author response image 1. sa3fig1:** (A) Immunohistochemical staining of a small intestinal cross-section from a Vil1^Cre^Rosa26^tdTomato/+^ Plp1^eGFP^ transgenic mouse in which enteric glia are labeled with green fluorescent protein (GFP) and intestinal epithelial cells are labeled with tdTomato. (B) Mucosal glia closely associate with epithelial cells in intestinal crypts. Scale bar – 20 µm.

The authors mention IL-22 as a possible link, but do Paneth cells express receptors for transmitters commonly released by enteric glia? Maybe they can have a look at putative cell-cell interactions by mapping ligand-receptor pairs in the scRNAseq datasets they used.

Beyond IL-22R, it is established that Paneth cells express receptors for secreted WNT proteins, which enteric glia have been shown to express (PMID: 34727519). This interaction could potentially be involved in glial regulation of Paneth cells, but mice lacking glia do not exhibit the same phenotypes as mouse models with disrupted WNT signaling. For example, animals lacking the WNT receptor Frizzled-5 in Paneth cells have mislocalization of Paneth cells to the villi (PMID: 15778706), which we do not readily observe in Plp1CreER-DTA mice. Furthermore, while mucosal enteric glia have been proposed as a source of WNT ligands, this role has been specifically attributed to GFAP+ cells, which may or may not be glia in the mucosa. Moreover, several other cell types in the mucosa around crypts have also been identified as significant sources of WNT ligands (PMID: 16083717, 22922422). We have now added these ideas to the Discussion.

Per the reviewer’s suggestion to use bioinformatics to explore other potential ligand-receptor pairings that might underlie glial regulation of Paneth cells, we conducted a CellPhoneDB analysis focused on these two cell types with a collaborator. This analysis highlighted a handful of potential ligand-receptor interactions, but none of these pathways could be clearly linked to the observed Paneth cell phenotype. Furthermore, virtually all the candidate interactions were not specific to glia, with the candidate ligands expressed by many other more abundant cell types in the mucosa. For these reasons, we decided not to include this analysis in the revised manuscript.

Previously the authors showed that enteric glia regulation of intestinal motility is sex-dependent (Rao et al 2017). While enteric glia depletion caused dysmotility in female mice, it did not affect motility in males. For this reason, most experiments in the current study were conducted in male mice only. However, for the experiments focusing on the effect of enteric glia depletion on hostmicrobiome interactions and intestinal microbiota composition both male and female mice were used. In Figure 8A male and female mice are distinctly depicted but this was not done for Figure 8C. Separate characterization of the microbiome of male and female mice would have helped to figure out how much intestinal dysmotility (in females) contributes to the effect on gut microbial composition. This is an important exercise to confirm that the effect on the microbiome is indeed a consequence of altered Paneth cell function, as suggested by the authors (in the results and discussion, and in the abstract).

In our microbiome analysis, we initially analyzed males and females separately but did not observe significant differences between the two sexes. Thus, we merged the data to increase the statistical power of the genotype comparisons. It was an oversight on our part to not label the datapoints by sex as we did for the other data in the manuscript. We have now revised the figures related to microbiome characterization (Fig. 5D-E and Supp. Fig. 8C) to indicate the sexes of the mice used. Stratifying the data by sex within-sample revealed no major sex-specific differences in microbiome diversity or enriched/depleted biomarkers in the core genotype-dependent observations.

In this context, it would also be interesting to compare the bulk sequencing data after enteric glia depletion between female and male mice.

Our bulk sequencing analysis of the effects of glial loss was conducted in male mice only in order to assess the effects independent of colonic dysmotility, a phenotype observed only in female Plp1CreER-DTA animals (PMID: 28711628). Given that we found rather muted transcriptional changes in male mice, we chose not to perform subsequent transcriptional analyses in female mice, further reasoning that any changes identified would most likely be attributable to dysmotility rather than direct glial effects. Future studies focusing on sex differences in the small intestine, where motility in the Plp1CreER-DTA model is unaffected by glial loss, could provide additional insights, especially in light of the recently reported sex differences in the gene expression and activity levels of enteric glia in the myenteric plexus (PMID: 34593632, 38895433).

**Reviewer #1 (Recommendations For The Authors):**
- Intro 2nd paragraph: please add to the sentence: "They found no major defects in epithelial properties AT STEADY STATE (or during homeostasis).

Revised as suggested.

- There seems to be a word missing in the 2nd sentence of paragraph 2 on page 4. "... but xxx consistent...".

Reviewed and there were no missing words.

- In the 2nd paragraph on page 8, when discussing GFAP expression in IBD patients, a reference is missing. Also, here it should be GFAP, not Gfap (in italics).

Revised as suggested.

**Reviewer #2 (Public Review):**
This is an excellent and timely study from the Rao lab investigating the interactions of enteric glia with the intestinal epithelium. Two early studies in the late 1990s and early 2000s had previously suggested that enteric glia play a pivotal role in control of the intestinal epithelial barrier, as their ablation using mouse models resulted in severe and fatal intestinal inflammation. However, it was later identified that these inflammatory effects could have been an indirect product of the transgenic mouse models used, rather than due to the depletion of enteric glia. In previous studies from this lab, the authors had identified expression of PLP1 in enteric glia, and its use in CRE driver lines to label and ablate enteric glia.In the current paper, the authors carefully examine the role of enteric glia by first identifying that PLP1-creERT2 is the most useful driver to direct enteric glial ablation, in terms of the number of glial cells targeted, their proximity to the intestinal epithelium, and the relevance for human studies (GFAP expression is rather limited in human samples in comparison). They examined gene expression changes in different regions of the intestine using bulk RNA-seq following ablation of enteric glia by driving expression of diphtheria toxin A (PLP1-creERT2;Rosa26-DTA). Alterations in gene expression were observed in different regions of the gut, with specific effects in different regions. Interestingly, while there were gene expression changes in the epithelium, there were limited changes to the proportions of different epithelial cell types identified using immunohistochemistry in control vs glial-ablated mice. The authors then focused on the investigation of Paneth cells in the ileum, identifying changes in the ultrastructural morphology and lysozyme activity. In addition, they identified alterations in gut microbiome diversity. As Paneth cells secrete antimicrobial peptides, the authors conclude that the changes in gut microbiome are due to enteric glia-mediated impacts on Paneth cell activity.Overall, the study is excellent and delves into the different possible mechanisms of action, including the investigation of changes in enteric cholinergic neurons innervating the intestinal crypts. The use of different CRE drivers to target enteric glial cells has led to varying results in the past, and the authors should be commended on how they address this in the Discussion.

We thank the reviewer for this positive feedback.

**Reviewer #2 (Recommendations For The Authors):**
I have a few minor comments:Changes in bacterial diversity - the authors make a very compelling case that changes in the proportions of various intestinal microbiome species were impacted by the change in Paneth cell secretions resulting from the depletion of enteric glia. Another potential mechanism of action could be alterations in gut motility resulting from loss of enteric glia. It appears that faecal samples were collected from both male and female mice, and hence changes in colonic motility could be involved. This should be addressed in the Results and Discussion.

We agree with the reviewer that GI dysmotility could influence microbial composition. To address this, we initially analyzed microbiome data separately for male and female mice, because only female Plp1CreER-Rosa26DTA exhibit dysmotility. We found no significant sex-specific differences in microbiome composition, however, which suggested to us that dysmotility was unlikely to be the primary driver of the observed microbial changes. Based on these findings, we opted to combine data from male and female mice in our final microbiome analysis. We have now revised the Results, Discussion, and Methods sections to clarify this.

Supplementary Figure 2: it would be helpful to include some labels of landmarks on the tissues, and arrows pointing to immunoreactive cells.

We have added labels and arrows to images in Supplementary Figure 2 per the reviewer’s suggestion.

Figure 4B: It's hard to tell the difference in ultrastructural morphology of the Paneth cells between Cre- and Cre+ mice in the EM images. Heterogeneous granules (PG) seem to be labelled in cells from both genotypes of mice. Some outlines of cells or arrows pointing to errant granules would be helpful.

We have added arrows indicated errant granules to images in Figure 4 per the reviewer’s suggestion.

**Reviewer #3 (Public Review):**
In this study, Prochera, et al. identify PLP1+ cells as the glia that most closely interact with the gut epithelium and show that genetic depletion of these PLP1+ glia in mice does not have major effects on the intestinal transcriptome or the cellular composition of the epithelium. Enteric glial loss, however, causes dysregulation of Paneth cell gene expression that is associated with morphological disruption of Paneth cells, diminished lysozyme secretion, and altered gut microbial composition.Overall, the authors need to first prove whether the Plp1CreER Rosa26DTA/+ mice system is viable.

In previous work, we discovered that the gene *Plp1* is broadly expressed by enteric glia and, within the mouse intestine, is quite specific to glial cells (PMID: 26119414). We characterized the Plp1CreER mouse line as a genetic tool in detail in this initial study. Then in a subsequent manuscript, we used Plp1CreER-DTA mice to genetically deplete enteric glia and study the consequences on epithelial barrier integrity, crypt cell proliferation, enteric neuronal health and gastrointestinal motility (PMID: 28711628). In this second study, we performed extensive validation of the Plp1CreER-DTA mouse model including detailed quantification of glial depletion in the small and large intestines across the myenteric, intramuscular and mucosa compartments by immunohistochemical (IHC) staining of whole tissue segments to sample thousands of cells. We found that the majority of S100B^+^enteric glia were depleted within 5 days in both sexes, including more than 88% loss of mucosal glia, and that this loss was stable at 3 subsequent timepoints (7, 9 and 14 days post-tamoxifen induction of Cre activity). Glial loss was further confirmed by IHC for GFAP in the myenteric plexus, and by ultrastructural analysis of the small intestine to ensure cell depletion rather than simply loss of marker expression. Our group was the first to use this model to study enteric glia, and since then similar models and our key observations have been replicated by other groups (PMID: 33282743, 34550727). Thus, we consider this model to be well established.

Also, most experimental systems have been evaluated by immunohistochemistry, scRNAseq, and electron microscopy, but need quantitative statistical processing.

RNA-sequencing and microbiome analyses are inherently quantitative (Figures 1A-B, Supp. Figure 1, Figure 2, Supp. Figure 4A, Figure 3A-B, Supp. Figure 5, Figure 5, and Supp. Figure 8C). Virtually all our other observations are also supported by quantitative analysis including analysis of mucosal glial markers (Fig. 1C-D), validation of Paneth cell transcript expression in the colon (Supp. Fig. 4B), measurement of epithelial cell type composition (Figure 3C, D), assessment of crypt innervation (Supp. Fig. 7E), and measurement of bacteria-to-crypt distance (Supp. Fig. 8A-B). The only observation that was not quantified was that of morphological abnormalities of Paneth cells. Given the inherently low sampling rate of EM studies, we felt that functional assays (explant secretion assays, effects on microbial composition) would be more meaningful for interrogation of a potential Paneth cell phenotype and thus elected to focus our quantitative analyses on those functional assays rather than further histological measurements.

In addition, the value of the paper would be enhanced if the significance of why the phenotype appeared in the large intestine rather than the small intestine when PLP1 is deficient for Paneth cells is clarified.

Please see detailed response to Reviewer 1 that addresses this comment and the corresponding addition to the Discussion.

Major Weaknesses:(1) Supplementary Figure 2; Cannot be evaluated without quantification.

Supplemental Figure 2 shows qualitative IHC observations that were highly reproducible across all the subjects indicated for each marker and align well with the quantitative transcriptional data from human subjects shown in Figure 1 and Supplemental Figure 1. The DAB staining in Supplemental Figure 2 could theoretically be quantified by staining intensity or counting cell number but we felt this would be arbitrary and difficult to achieve in a meaningful way with a single chromogen. The DAB reaction is associated with a non-linear relationship between amount of an antigen and staining intensity, especially at higher levels (PMID: 16978204, 19575836), because it is not a direct conjugate and relies upon an enzymatic reaction. The amplification step required for DAB staining using Horseradish Peroxidase (HRP) introduces variability, particularly with cytoplasmic markers and in complex tissue structures like the plexuses, where proteins are distributed throughout the glial network. Counting cell number also would not lead to fair comparisons between markers because while SOX10 shows a clear nuclear signal suitable for quantification, the other markers are all membrane or cytoplasmic proteins, making accurate counting nearly impossible in dense ganglia. Finally, quantifying cell number in 5-micron paraffin sections which have major differences in sampling from one subject to another in terms of presence of ganglia and ganglia size, would also make this prone to inaccuracy. Given these limitations and the robust qualitative data we have shown that aligns completely with the quantitative transcriptional analyses, we respectfully disagree with the reviewer’s comment.

(2) Figure 2A; Is Plp1CreER Rosa26DTA/+ mice system established correctly? S100B immunohistology picture is not clear. A similar study is needed for female Plp1CreER Rosa26DTA/+ mice. What is the justification for setting 5 dpt, 11 dpt? Any consideration of changes to organs other than the intestine? Wouldn't it be clearer to introduce Organoid technology?

Please see the detailed response to first comment. The Plp1CreER- DTA mouse model is well-established and there are detailed experimental justifications for the 5 and 11dpt timepoints as well as the focus on male mice for RNA-sequencing analyses. As described in our previous work (PMID: 28711628), Plp1^+^ cells throughout the animal would be affected, including Schwann cells and oligodendrocytes, which is why we limit our analyses to the first 11dpt, when there are fewer confounding variables. The S100B immunohistology picture in Figure 2A was intended to be a schematic graphical representation of the paradigm of glial loss, not a data figure. Extensive validation of glial loss in this model was shown in our previous study. To improve clarity, we have now enlarged the picture for the reader.

Regarding the suggestion to use organoid technology, standard intestinal epithelial organoids do not incorporate any elements of the enteric nervous system (ENS), which is the focus of this study. Some groups have made heroic efforts to incorporate ENS components into intestinal organoids by introducing neural crest progenitor cells and grafting the hybrid organoids under the renal capsule in mice (example PMID: 27869805); but these studies are still limited, and it remains unclear how much the preparations reflect functional, natively innervated intestine. Our ex vivo explant assay preserves native ENS-epithelial interactions, providing a more effective model for studying the relationship between enteric glia and Paneth cells.

(3) Figure 2B; Need an explanation for the 5 genes that were altered in the colon. Five genes should be evaluated by RT-qPCR. Why was there a lack of change in the duodenum and ileum?

While RT-qPCR validation of differentially expressed genes was once common practice, especially with microarray data, there is now robust evidence for strong correlations between RNA sequencing (RNAseq) results and RT-qPCR measurements of gene expression (PMID: 26208977, 28484260). Notably Rajkumar et al. (PMID: 26208977) demonstrated that RNAseq analyzed using DESeq2 (a method which we employed in our study), yields highly accurate results. They reported a 0% false positive rate and a 100% positive predictive value for DESeq2, rendering additional RT-qPCR validation redundant. We only performed RT-qPCR analysis of colonic *Lyz1* expression because our IHC analyses failed to show any ectopic expression of the protein in the colons of Cre^+^ mice (Supp. Figure 4D) and we wished to validate the gene expression change seen by RNAseq in an independent cohort to be absolutely sure. Per the detailed response to Reviewer 1, we do not have a mechanistic explanation for why there is selective transcriptional induction of Paneth cell-related genes in the colon upon glial depletion. We have elaborated on this in the revised Discussion.

(4) Supplementary Figure 3; Top 3 genes should be evaluated by RT-qPCR.

Given that none of the changes included in Supplementary Figure 3 for the duodenum or ileum reach the standard threshold for statistical significance and in view of the findings by Rajkumar, et al. (2015) described above, we don’t believe that evaluating expression of these genes by RT-qPCR would be informative in interpreting these negative results.

(5) Supplementary Figure 4B, C, and D; Why not show analysis in the small intestine?

We chose to focus on the colon for this analysis because this was the only region of the intestine that exhibited statistically significant differences in transcriptional profiles as assessed by DEG.

(6) Supplementary Figure 4D; Cannot be evaluated without quantification.

As shown in the representative images, no LYZ1 or DEFA5 signal was detected in the colons of Cre^-^ or Cre^+^ mice (n=3 mice per genotype; >100 crypts/mouse assessed), though it was readily detectable in the ileums of both genotypes. We have now added the number of crypts assessed to the figure legend.

(7) Figure 3D; Cannot be evaluated without quantification.

Please see Fig. 3C for quantification of each cell type marker shown in Figure 3D.

(8) Supplementary Figure 5B and C; Top 3 genes should be evaluated by RT-qPCR.

Please see detailed explanation to comments #3 and #4 above.

(9) Supplementary Figure 6; Top 3 genes should be evaluated by RT-qPCR.

This comment was likely made in error because Supplementary Fig. 6 does not show any gene expression data.

(10) Figure 4A; Cannot be evaluated without quantification.

We appreciate the reviewer’s comment here and strived very hard to add quantification of the Paneth cell granule phenotype seen by light microscopy to our study. IHC for LYZ1 is typically the gold standard for assessment of Paneth cell granules by light microscopy. In our hands, however, we encountered persistent issues with IHC for this protein. While it very reproducibly detected Paneth cells with sufficient specificity to enable quantification of number of immunoreactive cells (as shown in Figure 3C), it did not enable quantification of granule morphology because it consistently exhibited diffuse staining throughout the cell (see Author response image 2 below). This appearance persisted regardless of extensive titration of fixation parameters (time, temperature, fixative supplier, 10% NBF vs 4% PFA), tissue preparation (fixed as intact tubes versus “swiss-rolls”), permeabilization conditions, operator, antibody used, and other variables. Upon subsequently surveying the literature, it seems that similar diffuse staining patterns for LYZ1 have been observed by numerous other groups and this may simply be an experimental limitation.

**Author response image 2. sa3fig2:** Representative IHC images showing LYZ1 staining optimization. Ileal tissues from 8-10-week-old mice were prepared as either 'swiss-rolls' (A-D) or tubes (E, F) and fixed using different protocols: 10% neutral buffered formalin (NBF) from Epredia (#5710-LP) (A-B, E), 10% NBF from G-Biosciences (#786-1057) (C, D), or 4% paraformaldehyde (PFA) from VWR (#100503-917) (F). Fixations were conducted at room temperature (A, C) or at 4°C (B, D-F). Diffuse cytoplasmic LYZ1 staining is observed within Paneth cells, regardless of conditions of tissue preparation.

As an alternative approach to detecting Paneth cell granules, we tried UEA-I lectin staining. This labeling approach was sufficient to reveal qualitative differences in Paneth granule morphology in Cre^+^ mice, as shown in Fig. 4A. However, the transient nature of this lectin labeling made it very difficult to systematically quantify granule morphology in a blinded manner, as we did for our other analyses. Given these persistent challenges, we decided to present qualitative data on morphology by two orthogonal approaches (UEA-I staining by light microscopy and ultrastructure by EM) and focus on functional read-outs for quantitative analyses (explant secretion assays and microbiome analyses). In aggregate, we feel that these data provide robust and complementary evidence of the observed phenotype from independent experimental approaches.

(11) Figure 4D; Cannot be evaluated without quantification.

This comment was likely made in error because there is no Figure 4D.

(12) Additional experiments on in vivo infection systems comparing Plp1CreER Rosa26DTA/+ mice and controls would be great.

We agree that in vivo infection experiments would be very interesting to pursue, particularly given the potential role of Paneth cells in innate immunity. These studies are beyond the scope of the current manuscript, but we hope to report on them in the future.

**Reviewer #3 (Recommendations For The Authors)**:Patients with inflammatory bowel disease (IBD); UC or CD.

Revised per reviewer suggestion.